# GAUSSIAN MUTUAL INFORMATION MAXIMIZATION FOR GRAPH SELF-SUPERVISED LEARNING: BRIDGING CONTRASTIVE-BASED TO DECORRELATION-BASED

## ABSTRACT

Enlightened by the *InfoMax* principle, graph contrastive learning has achieved remarkable performance in processing large amounts of unlabeled graph data. Due to the impracticality of precisely calculating mutual information (MI), conventional contrastive methods turn to approximate its lower bound using parametric neural estimators, which inevitably introduces additional parameters and leads to increased computational complexity. Building upon a common Gaussian assumption on the distribution of node representations, a computationally tractable surrogate for the original MI can be rigorously derived, termed as Gaussian Mutual Information (GMI). GMI eliminates the reliance on parameterized estimators and negative samples, resulting in an efficient contrastive objective with provable performance guarantees. Another parallel research branch on decorrelation-based self-supervised methods has also emerged, with the core idea of mitigating dimensional collapse by decoupling various representation channels. While the differences between the two families of contrastive-based and decorrelation-based methods have been extensively discussed to inspire new approaches, their potential connections are still obscured in the mist. By positioning the proposed GMI-based objective with cross-view identity constraint as a pivot, we bridge the gap between these two research areas from two aspects of approximate form and consistent solution, which contributes to the advancement of a unified theoretical framework for self-supervised learning. Extensive comparison experiments and visual analysis provide compelling evidence for the effectiveness and efficiency of our method while supporting our theoretical achievements. Besides, the empirical evidence indicates that even in cases deviating from Gaussianity, our approach continues to maintain its performance, which significantly extends application scenarios.

## 1 INTRODUCTION

The scarcity of task-related annotations for graph data, which usually rely on domain knowledge and specific equipment such as chemical instruments (Liu et al., 2023), urgently calls for the emergence of advanced unsupervised learning methods without manual supervision. In this context, graph self-supervised learning (SSL) (Zhu et al., 2020; 2021; You et al., 2020; Thakoor et al., 2022; Bielak et al., 2021) arises naturally in response to the prevailing demand, approaching and even surpassing the performance of their supervised counterparts (Kipf & Welling, 2016a; Veličković et al., 2017; Hamilton et al., 2017). SSL models are trained on well-designed pretext tasks in a task-agnostic manner, whose optimization results in general, meaningful, and transferable representations for downstream applications. As a distinguished member of the SSL family, multi-view learning with Siamese networks (Li et al., 2022) has demonstrated exceptional performance and garnered widespread interest. At the heart of such methods is to extract invariant or common information from various augmented views of the same instance (*i.e.*, positive pairs) while adopting specific strategies to prevent model collapse. The existing multi-view learning methods can be classified into two distinct categories based on their means of addressing degradation: *contrastive* (Zhu et al., 2020; 2021; You et al., 2020; Hassani & Khasahmadi, 2020) and *non-contrastive* (Thakoor et al., 2022; Bielak et al., 2021; Zhang et al., 2021) approaches. The former suppresses encoded representations from collapsing into a constant point by pushing negative pairs apart, while the latter employs special

strategies such as decorrelating different representation dimensions (Zhang et al., 2021; Bielak et al., 2021) or designing asymmetric network architecture (Thakoor et al., 2022; Grill et al., 2020).

The concept of *contrastive* multi-view learning originates from information theory (Ash, 2012; Walters-Williams & Li, 2009), aiming to improve the consistency between various views by maximizing their mutual information (MI). Nevertheless, the exact computation of mutual information for high-dimensional continuous variables is usually intractable. To cope with this challenge, some previous endeavors have attempted to employ parameterized neural estimators to perform an empirical evaluation of mutual information from finite samples, yielding notable achievements like MINE (Belghazi et al., 2018), Jensen-Shannon estimator (Nowozin et al., 2016), and *InfoNCE* (Gutmann & Hyvärinen, 2010a). Formally, contrastive learning methods equipped with the parameterized estimator of MI manifest as a contrastiveness between positive pairs from a joint distribution and negative pairs from two marginal ones. Despite their decent performance, these methods are accompanied by several inherent drawbacks: a) a substantial number of samples are required to obtain reliable estimation and achieve satisfactory results, which inevitably increases computational burden; b) the incorporation of parameterized MI estimators amplifies the complexity of SSL models.

Deviating from the conventional graph contrastive learning methods, we delve into lightweight and efficient alternatives with no reliance on parameterized MI estimators for node-level representation learning. Assuming node representations obey a Gaussian distribution, a feasible closed-form solution can be obtained, called Gaussian Mutual Information (GMI), through tractable integration operations on the native definition of MI. In its mathematical form, GMI exclusively depends on the covariance matrices. As a result, the estimation of MI is alleviated to that of covariance matrices, which can be effortlessly obtained from empirical data (*i.e.*, node representations). Independent of additional architecture, the resultant objective under GMI can be directly calculated within the representation space, leading to higher computational efficiency and better resource friendliness. Most importantly, the performance of the proposed method can still hold even when actual scenarios deviate from Gaussian distributions, thereby extending its applicability beyond Gaussian constraints.

As another indispensable branch of the SSL family, the decorrelation-based non-contrastive methods (Zhang et al., 2021; Bielak et al., 2021; Bardes et al., 2022; Ermolov et al., 2021) prevent degenerate solutions and learn diverse representations by decoupling various channels, whose objective functions exhibit an utterly distinct appearance from those of contrastive-based ones. While the distinctions between the two branches have been thoroughly discussed, their latent theoretical relationships remain enshrouded in ambiguity. Imposing a cross-view identity constraint, which enhances the perfect alignment of representations from different views of the same instance, to our proposed GMI-based objective function, we employ the newly induced objective as a pivot to elucidate the underlying connections between decorrelation-based and contrastive-based methods. On the one hand, the former is formally equivalent to a second-order Taylor series expansion of the latter. On the other hand, their objectives share consistent solutions. Overall, the decorrelation-based methods can be regarded as an instantiation of contrastive learning under the Gaussian assumption on the distribution of node representations and identity constraint.

Our contributions in this paper are summarized as follows:

- Under the common Gaussian assumption for node representations, we propose a computationally tractable graph contrastive objective based on mutual information maximization. Due to its simplicity and lightweight nature, it exhibits high efficiency with provable performance guarantees.
- We bridge decorrelation-based self-supervised methods to our proposed contrastive objective from two aspects of approximation of form and consistency of solution, which points out a clue to demystify the relationships between various self-supervised learning methods.
- Extensive empirical studies demonstrate the effectiveness and efficiency of our method compared with state-of-the-art peers. Additionally, exploratory studies and visual analysis further reveal the advantages of our method and reinforce the understanding of our theoretical achievements.

## 2 RELATED WORK

### 2.1 GRAPH SELF-SUPERVISED LEARNING

For its remarkable performance, multi-view-based methods have been the dominant paradigm of graph self-supervised learning, which expect to explore common information from various augmented

versions. A crucial aspect of these methods is to prevent degenerate solutions, where all representations are collapsed to a constant point (*i.e.*, complete collapse) or a subspace (*i.e.*, dimensional collapse) of the entire representation space. The current methods can be categorized into two groups, namely contrastive (Zhu et al., 2020; 2021; You et al., 2020; Hassani & Khasahmadi, 2020; Peng et al., 2020) and non-contrastive (Thakoor et al., 2022; Bielak et al., 2021; Zhang et al., 2021) approaches, based on their ways to circumvent model collapse.

The contrastive-based methods usually follow the criterion of mutual information maximization (Hjelm et al., 2019; Linsker, 1988), whose objective functions take the form of contrasting positive pairs with negative ones. As pioneer works, Deep Graph Infomax (DGI) (Veličković et al., 2018) and InfoGraph (Sun et al., 2020) learn unsupervised representations by maximizing mutual information between node-level representations and a whole graph summary vector based on the Jenson-Shannon estimator (Nowozin et al., 2016). GraphCL (You et al., 2020), GRACE (Zhu et al., 2020), and GCA (Zhu et al., 2021) embed the *InfoNCE* (Gutmann & Hyvärinen, 2010b) loss into graph contrastive learning framework. From the view of information theory, InfoGCL (Xu et al., 2021a) investigates how to build appropriate contrastive learning frameworks for specific tasks.

The non-contrastive methods discard negative samples, which require special strategies to avoid collapsed solutions. BGRL (Thakoor et al., 2022) utilizes asymmetric architecture and a stop-gradient strategy to prevent the two branches from merging. Graph Barlow Twins (G-BT) (Bielak et al., 2021) generalizes the celebrated Barlow Twins (Zbontar et al., 2021) from images to graph data. CCA-SSG (Zhang et al., 2021) learns augmentation-invariant information while decorrelating features in different dimensions to prevent degenerated solutions.

## 2.2 ESTIMATING MUTUAL INFORMATION

Mutual information is a powerful and commonly used measure for general correlation between random variables, which has been applied to a range of fields, including medical image processing (Pluim et al., 2003), feature selection (Estevez et al., 2009; LIU, 2009), information bottleneck (Goldfeld & Polyanskiy, 2020), and recommendation system (Sankar et al., 2020). Nevertheless, the exact computation of MI for high-dimensional variables is notoriously difficult. An alternative scheme is to estimate MI from empirical observations.

The non-parametric estimators make no assumptions about the underlying distribution of data and require no specification of any parameters. The most popular class in this branch is the k-nearest-neighbor-based estimators and their extensions (Singh et al., 2003; Kraskov et al., 2004; Gao et al., 2015). Besides, the methods based on kernel density estimation (KDE) first estimate the probability density function and then compute MI by Monte-Carlo integration (Silverman, 1986; Scott, 2015).

The research on neural-network-based MI estimation (Belghazi et al., 2018; Nowozin et al., 2016; Gutmann & Hyvärinen, 2010a) has also made significant process, which has been widely applied in representation learning. The key technical ingredient of these methods is to approximate the lower bound of MI based on dual representations of the $f$-divergence (Nowozin et al., 2016).

## 3 METHODOLOGY

### 3.1 PRELIMINARIES AND OVERALL FRAMEWORK

**Preliminaries.** Before further discussion, the preliminary conceptions presented in this paper are first provided. A graph is denoted by $G(\mathbf{A}, \mathbf{X})$ with node set $\mathcal{V} = \{v_1, ..., v_N\}$ and edge set $\mathcal{E}$, where $|\mathcal{V}| = N$ indicates the number of nodes. Each node $v_i \in \mathcal{V}$ has a $D$-dimensional feature vector $\mathbf{x}_i \in \mathbb{R}^D$. Node feature matrix $\mathbf{X} = [\mathbf{x}_1, ..., \mathbf{x}_N]^\top \in \mathbb{R}^{N \times D}$ contains feature information of all nodes and adjacency matrix $\mathbf{A} \in \mathbb{R}^{N \times N}$ describes the connection relationship between nodes. The task of node-level graph self-supervised learning is to seek good node representations $\widetilde{\mathbf{H}} = [\tilde{\mathbf{h}}_1, ..., \tilde{\mathbf{h}}_N]^\top \in \mathbb{R}^{N \times d}$ through learning a continuous mapping $f_\theta(\mathbf{A}, \mathbf{X}) : \mathbb{R}^{N \times N} \times \mathbb{R}^{N \times D} \to \mathbb{R}^{N \times d}$ without labels, where $\theta$ denotes learnable parameters and $d$ indicates the representation dimension.

**Graph View Generation.** Let the transformation $\tau \in \mathcal{T} : G(\mathbf{A}, \mathbf{X}) \to G'(\mathbf{A}', \mathbf{X}')$ map the original graph to an augmented version, where $\mathcal{T}$ denotes the whole function space for augmentation. Specifically, the graph augmentation $\tau$ is jointly implemented from two aspects of graph topology and

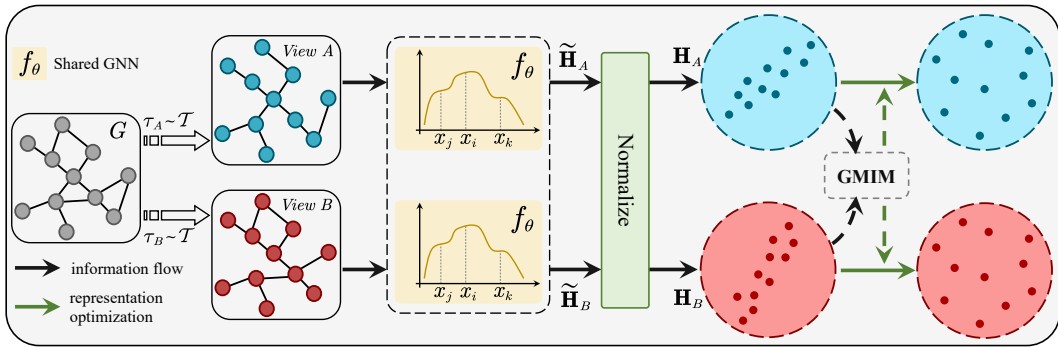

Figure 1: An overview of the overall framework based on GMIM. The outputs of well-trained $f_\theta(\cdot)$ can be applied to various node-level downstream tasks. Best viewed in colors.

feature, following previous works (Zhu et al., 2020). For topology-level augmentation, *edge removal* is adopted, randomly removing edges of a certain ratio $p_e$ on the original graph. For feature-level augmentation, *node feature masking* randomly sets feature channels of a specific number $D \cdot p_f$ in feature matrix $\mathbf{X} \in \mathbb{R}^{N \times D}$ to zero, where $p_f$ is the masking ratio.

**Overall Framework.** In terms of basic framework, this paper inherits the common practice of prior studies. As shown in Figure 1, two various views $G'_A(\mathbf{A}'_A, \mathbf{X}'_A) = \tau_A(G)$ and $G'_B(\mathbf{A}'_B, \mathbf{X}'_B) = \tau_B(G)$ are firstly generated based on two graph augmentation functions $\tau_A$ and $\tau_B$ randomly sampled from $\mathcal{T}$. The two augmented versions are fed into a shared graph convolutional network (Kipf & Welling, 2016a) $f_\theta(\cdot)$ to obtain representations $\widetilde{\mathbf{H}}_A = [\tilde{\mathbf{h}}_1^A, ..., \tilde{\mathbf{h}}_N^A]^\top$ and $\widetilde{\mathbf{H}}_B = [\tilde{\mathbf{h}}_1^B, ..., \tilde{\mathbf{h}}_N^B]^\top$. To facilitate subsequent discussion, $\widetilde{\mathbf{H}}_A$ and $\widetilde{\mathbf{H}}_B$ are further batch-normalized into $\mathbf{H}_A = [\mathbf{h}_1^A, ..., \mathbf{h}_N^A]^\top$ and $\mathbf{H}_B = [\mathbf{h}_1^B, ..., \mathbf{h}_N^B]^\top$, each representation channel in which obey a distribution with 0-mean and 1-standard deviation. "GMIM" is the optimization objective proposed in the following sections.

## 3.2 GAUSSIAN MUTUAL INFORMATION MAXIMIZATION

Contrastive learning is initially enlightened by the *InfoMax* principle (Bell & Sejnowski, 1995), which expects to maximize mutual information between representations from various views.

**Definition 1** (Mutual Information). *Let $X$ and $Y$ denote two d-dimensional continuous variables with marginal probability functions $p_x(X)$ and $p_y(Y)$, respectively. Their joint probability density is indicated by $p_{x,y}(X, Y)$. The mutual information $I(X; Y)$ between $X$ and $Y$ is defined as*

$$I(X; Y) = \int_{\mathcal{X}} \int_{\mathcal{Y}} p_{x,y}(X, Y) \ln \frac{p_{x,y}(X, Y)}{p_x(X) \cdot p_y(Y)} dX dY, \tag{1}$$

*where $\mathcal{X}$ and $\mathcal{Y}$ denote domains corresponding to $X$ and $Y$, respectively.*

Nevertheless, the exact computation of mutual information for high-dimensional continuous variables is usually infeasible. First, it is challenging to estimate the probability densities from empirical observations. Second, even though they can be obtained, which may have complex forms, the integral operation in Eq. (1) remains difficult, even intractable. To tackle these issues, the conventional contrastive leaning methods employ parametric networks to directly estimate a lower bound of MI, which can be trained alongside the backbone via back-propagation in an end-to-end manner.

Divergent from the peer works, this paper assumes a latent Gaussian distribution for node representations and drops parametric estimators, which leads to a computationally tractable surrogate. The Gaussian assumption is justifiable and extensively employed in numerous disciplines to simplify analysis and calculation, including economics, data science, and physics (Park et al., 2013).

**Proposition 1** (Gaussian Mutual Information). *If the variables $X$ and $Y$ obey two multi-dimensional Gaussian distributions, respectively, the Gaussian mutual information $I_G(X; Y)$ between them is*

$$I_G(X; Y) = \frac{1}{2} \ln \frac{\det(\mathbf{\Sigma}_X) \cdot \det(\mathbf{\Sigma}_Y)}{\det(\mathbf{\Sigma}_{X,Y})}, \tag{2}$$

*where* $\det(\cdot)$ *indicates the determinant of a matrix,* $\mathbf{\Sigma}_X$ *and* $\mathbf{\Sigma}_Y$ *are the covariance matrices of* $X$ *and* $Y$, *respectively, and* $\mathbf{\Sigma}_{X,Y} = \begin{bmatrix} \mathbf{\Sigma}_X & \mathbf{\Sigma}_{XY} \\ \mathbf{\Sigma}_{XY}^\top & \mathbf{\Sigma}_Y \end{bmatrix}$ *is the covariance matrix of variable* $[X^\top, Y^\top]^\top$ *with the cross-covariance matrix* $\mathbf{\Sigma}_{XY}$.

*Proof.* Please refer to Appendix A. □

The three covariance matrices $\mathbf{\Sigma}_X$, $\mathbf{\Sigma}_Y$, and $\mathbf{\Sigma}_{X,Y}$ can be effortlessly estimated from the empirical data, which results in a straightforward calculation of Gaussian mutual information. The covariance matrix is a real symmetric matrix whose eigenvalues are all *greater than or equal to zero*. Mathematically, the determinant of a matrix is numerically equal to the product of all eigenvalues. Due to the underlying dimensional collapse issue during self-supervised pretraining, many eigenvalues of the empirical covariance matrix tend to be zero, which causes its determinant to approach zero. Therefore, a direct adoption of Eq. (2) for constructing a contrastive learning objective will bring about numerical instability. One feasible strategy to alleviate the numerical issue is to offset and scale the eigenvalues of the matrix performed by $\det(\cdot)$. As a result, a practical objective based on Gaussian Mutual Information Maximization (GMIM) can be formulated as

$$\mathcal{L}_{\text{GMIM}} = \ln \frac{\det(\mathbf{I} + \eta \cdot \mathbf{\Sigma}_{A,B})}{\det(\mathbf{I} + \eta \cdot \mathbf{\Sigma}_A) \cdot \det(\mathbf{I} + \eta \cdot \mathbf{\Sigma}_B)}, \tag{3}$$

where $\mathbf{\Sigma}_A = \frac{1}{N}\mathbf{H}_A^\top\mathbf{H}_A$, $\mathbf{\Sigma}_B = \frac{1}{N}\mathbf{H}_B^\top\mathbf{H}_B$, $\mathbf{\Sigma}_{A,B} = \frac{1}{N}\begin{bmatrix} \mathbf{H}_A^\top\mathbf{H}_A & \mathbf{H}_A^\top\mathbf{H}_B \\ \mathbf{H}_B^\top\mathbf{H}_A & \mathbf{H}_B^\top\mathbf{H}_B \end{bmatrix}$, $\mathbf{I}$ is an identity matrix, and $\eta$ is a scaling factor with a typical value of $0.1$. The eigenvalues of $\mathbf{I} + \eta \cdot \mathbf{\Sigma}_A$ fall into $[1, +\infty)$, and so do the other two sibling matrices.

According to (Cover, 1999), the following property holds:

**Property 1.** *For variables* $X$ *and* $Y$, *the relationship between entropy and mutual information is*

$$I(X;Y) = H(X) - H(X|Y), \tag{4}$$

*where* $H(X) = -\int_{\mathcal{X}} p_x(X) \ln p_x(X) dX$ *denote the information entropy of* $X$ *under* $p_x(X)$, *and* $H(X|Y) = \int_{\mathcal{X}} \int_{\mathcal{Y}} p_{x,y}(X,Y) \ln \frac{p_{x,y}(X,Y)}{p_y(Y)} dX dY$ *is the conditional entropy of* $X$ *given* $Y$. *If* $X$ *is deterministic given* $Y$, $H(X|Y) = 0$. *Symmetrically,* $I(X,Y) = H(Y) - H(Y|X)$ *holds.*

From Property 1, it can be known that mutual information maximization actually involves two potential processes: increasing information entropy and reducing conditional entropy. The conditional entropy is minimized when the relationship between $X$ and $Y$ can be described by a deterministic function $g(\cdot)$, that is, $Y' = g(X')$ holds for any pair $(X', Y') \sim p_{x,y}$. In our setup of overall framework, a shared graph neural network is employed, expecting that representations of different versions from the same instance can match each other perfectly. In this circumstance, $g(\cdot)$ is preferred to be an identity mapping. By imposing the cross-view identity constraint to mutual information maximization with the preservation of entropy maximization, we can obtain an objective under Gaussian Mutual Information Maximization with Identity Constraint (GMIM-IC):

$$\mathcal{L}_{\text{GMIM-IC}} = \underbrace{\frac{1}{N}\sum_{v \in \mathcal{V}} \|\mathbf{h}_v^A - \mathbf{h}_v^B\|_2^2}_{\text{identity constraint}} - \beta \cdot \underbrace{\left( \ln\det(\mathbf{I} + \eta \cdot \mathbf{\Sigma}_A) + \ln\det(\mathbf{I} + \eta \cdot \mathbf{\Sigma}_B) \right)}_{\text{entropy maximization}}, \tag{5}$$

where $\beta$ is a coefficient balancing identity constraint term and entropy maximization term. Some analysis about Eq. (5) is placed in Appendix B.

## 4 BRIDGING CONTRASTIVE-BASED TO DECORRELATION-BASED

Based on the symbols in this article, the decorrelation-based self-supervised method (taking CCA-SSG (Zhang et al., 2021) as an example) can be formularized as

$$\mathcal{L}_{\text{CCA-SSG}} = \underbrace{\frac{1}{N}\|\mathbf{H}_A - \mathbf{H}_B\|_F^2}_{\text{invariance term}} + \lambda \cdot \underbrace{\left( \|\mathbf{\Sigma}_A - \mathbf{I}\|_F^2 + \|\mathbf{\Sigma}_B - \mathbf{I}\|_F^2 \right)}_{\text{decorrelation term}}, \tag{6}$$

where $\lambda$ denotes a balancing factor and $\|\cdot\|_F$ indicates the Frobenius norm of a matrix. Since the diagonal elements of $\mathbf{\Sigma}_A$ are always 1, the following equation holds:

$$\|\mathbf{\Sigma}_A - \mathbf{I}\|_F^2 = \sum_{i=1}^{d} \sum_{j=1,j\neq i}^{d} (\Sigma_A^{ij})^2, \tag{7}$$

where $\Sigma_A^{ij}$ represents the element in the $i$-th row and the $j$-th column of $\mathbf{\Sigma}_A$. The conclusion in Eq. (7) still holds for $B$. Next, we will establish connections between the decorrelation-based methods and our objective $\mathcal{L}_{\texttt{GMIM-IC}}$ from two perspectives.

## 4.1 EXPLAINATION 1: APPROXIMATE FORM

**Lemma 1.** *For a square matrix* $\mathbf{M}$, $\det(\exp(\mathbf{M})) = \exp(\mathrm{tr}(\mathbf{M}))$. *Replace* $\mathbf{M}$ *with* $\ln(\mathbf{I} + \eta \cdot \mathbf{\Sigma}_*)$ :

$$\ln \det(\mathbf{I} + \eta \cdot \mathbf{\Sigma}_*) = \mathrm{tr}(\ln(\mathbf{I} + \eta \cdot \mathbf{\Sigma}_*)), \tag{8}$$

*where* $* \in \{A, B\}$ [1]. *Applying Taylor expression to the logarithmic function in* $\mathrm{tr}(\ln(\mathbf{I} + \eta \cdot \mathbf{\Sigma}_*))$, *it can be known that*

$$\ln \det(\mathbf{I} + \eta \cdot \mathbf{\Sigma}_*) = tr \left( \sum_{k=1}^{+\infty} \frac{(-1)^{k+1}}{k} (\eta \cdot \mathbf{\Sigma}_*)^k \right). \tag{9}$$

Based on Lemma 1, we can obtain a second-order Taylor approximation:

$$- \ln \det(\mathbf{I} + \eta \cdot \mathbf{\Sigma}_A) \approx \frac{\eta^2}{2} \cdot \sum_{i=1}^{d} \sum_{j=1,j\neq i}^{d} (\Sigma_A^{ij})^2 + \frac{\eta^2}{2} \cdot d - \eta \cdot d. \tag{10}$$

The proof of Lemma 1 and detailed derivations of Eq. (10) are placed in Appendix C.1.

Comparing Eq. (7) with Eq. (10), $\|\mathbf{\Sigma}_A - \mathbf{I}\|_F^2$ is equivalent to the second-order Taylor expression of $-\ln \det(\mathbf{I} + \eta \cdot \mathbf{\Sigma}_A)$ without considering the constant term. Symmetrically, the finding can be extended to view $B$. Besides, the invariance term in Eq. (6) has an identical form with the identity constraint term in Eq. (5). Thus, we can conclude that the objective of decorrelation-based methods such as CCA-SSG has a approximate form with that of GMIM-IC.

## 4.2 EXPLAINATION 2: CONSISTENT SOLUTION

Certainly, the objective function in Eq. (6) is minimized when the representations from the two views are perfectly matched and their empirical covariance matrices tend towards the identity matrix.

**Proposition 2.** *When* $\ln \det(\mathbf{I} + \eta \cdot \mathbf{\Sigma}_*)$ *or* $\ln \det(\mathbf{\Sigma}_*)$ *is maximized, the empirical covariance matrix* $\mathbf{\Sigma}_*$ *will converge to an identity matrix.*

*Proof.* Refer to Appendix C.2. $\qquad\qquad\square$

Obviously, the identity constraint term is minimized in Eq. (5) when $\mathbf{H}_A$ and $\mathbf{H}_B$ is completely aligned. Combining this observation with Proposition 2, it can be concluded that the decorrelation-based objective in Eq. (6) has the same solution as the objective based on GMIM-IC.

Explaination 1 and 2 demonstrate the relationship between two objectives $\mathcal{L}_{\texttt{CCA-SSG}}$ and $\mathcal{L}_{\texttt{GMIM-IC}}$ from two aspects of approximation in form and consistency in final solutions. Consequently, the following corollary emerges naturally.

**Corollary 1.** *The decorrelation-based graph self-supervised methods, which expect to align multiple views and disentangle different representation dimensions, can actually be viewed as a special instance of mutual-information-maximization-based contrastive learning under the Gaussian assumption and identity constraint.*

---

[1] In the remaining sections of this article, $*$ is used to represent either $A$ or $B$.

## 5 THEORETICAL ANALYSIS

### 5.1 PREVENTING DIMENSIONAL COLLAPSE

When dimensional collapse issue exists, various representation channels are coupled to each other and present a certain correlation. Another manifestation of dimensional collapse is that data points exhibit differences in distributions along different principal directions, where some directions exhibit loose distributions with higher variance, while others present tight distributions with lower variance.

**Property 2.** *For empirical covariance matrix $\mathbf{\Sigma} = \frac{1}{N}\mathbf{H}^\top\mathbf{H} \in \mathbb{R}^{d \times d}$ with batch-normalized representations $\mathbf{H} = [\mathbf{h}_1, ..., \mathbf{h}_N]^\top \in \mathbb{R}^{N \times d}$, which has $d$ eigenvalues $[\lambda_1, \lambda_2, \ldots, \lambda_d]$ corresponding to $d$ eigenvectors $[\mathbf{q}_1, \mathbf{q}_2, \ldots, \mathbf{q}_d]$, the variance of data $\mathbf{H}$ along the $k$-th principal direction (that is, the direction of $\mathbf{q}_k$) is numerically equal to $\lambda_k$.*

*Proof.* Please refer to Appendix C.3. □

Property 2 potentially suggests that the unevenness of the eigenvalues of the covariance matrix leads to the issue of dimensional collapse. Combining with Proposition 2, it can be known that maximizing the logarithm of determinant can ensure entropy maximization and realize isotropic covariance, which actually guarantees the evenness of eigenvalues of the covariance matrix and thus prevents dimensional collapse issue. From the perspective of representation learning, this result will enhance the diversity, richness, and discriminability of node representations, thereby conferring advantages to downstream tasks.

### 5.2 RELATION WITH *InfoNCE*

As the commonest indicator in contrastive learning, the *InfoNCE* loss guides the model to learn meaningful and diverse representations by pulling together embeddings from positive pairs and pushing apart those from negative ones on the unit hypersphere.

A previous work (Wang & Isola, 2020) decomposes the classical *InfoNCE* objective into two terms: alignment term and uniformity term. The alignment term expects to match two views, which shares the same purpose as our identity constraint. The uniformity term is utilized to distribute representations uniformly on the unit hypersphere $\mathcal{S}^{d-1}$.

**Proposition 3.** *When the representations scatter over the unit hypersphere $\mathcal{S}^{d-1}$ uniformly (that is, they obey a complete uniform distribution), their entropy will reach the maximum value.*

*Proof.* Please refer to Appendix C.4. □

Proposition 3 suggests that the uniformity term implicitly realize the maximization of entropy by distributing the representations uniformly over the hypersphere. In contrast, our method explicitly maximizes the entropy of representations under the assumption of Gaussian distribution. In general, the two approaches reach the similar goal by different routes.

## 6 EXPERIMENTS

### 6.1 DATASETS AND EXPERIMENTAL SETUP

**Datasets.** To assess our approach, six widely used benchmark datasets are adopted for experimental study, including three citation networks **Cora**, **Citeseer**, and **Pubmed** (Sen et al., 2008), two co-purchase networks **Amazon-Computers** and **Amazon-Photo** (Shchur et al., 2019), and one co-authorship network **Coauthor-CS** (Shchur et al., 2019).

**Experimental Setup.** The representation encoder is implemented by Graph Convolutional Network (GCN) (Kipf & Welling, 2016a). The model parameters are initialized via Xavier initialization (Glorot & Bengio, 2010) and trained by Adam optimizer (Kingma & Ba, 2017). All experiments are conducted on a TITAN RTX GPU with 24 GB memory. The representations are first learned by our method in an unsupervised manner and then evaluated by a simple linear classifier.

Table 1: Node classification accuracy with standard deviation in percentage on six datasets. The "**Input**" column illustrates the data used in the training stage, and $\mathbf{Y}$ denotes labels. The **bold** font highlights the best results. "OOM" means Out-Of-Memory.

| | Algorithm | Input | Cora | Citeseer | Pubmed | Computers | Photo | Coauthor-CS |
|---|---|---|---|---|---|---|---|---|
| | MLP | $\mathbf{X, Y}$ | $57.8 \pm 0.2$ | $54.2 \pm 0.1$ | $72.8 \pm 0.2$ | $79.81 \pm 0.06$ | $86.36 \pm 0.08$ | $91.32 \pm 0.11$ |
| | GCN | $\mathbf{X, A, Y}$ | $81.5$ | $70.3$ | $79.0$ | $86.51 \pm 0.54$ | $92.42 \pm 0.22$ | $93.03 \pm 0.31$ |
| | GAT | $\mathbf{X, A, Y}$ | $83.0 \pm 0.7$ | $72.5 \pm 0.7$ | $79.0 \pm 0.3$ | $86.93 \pm 0.29$ | $92.56 \pm 0.35$ | $92.31 \pm 0.24$ |
| Unsupervised | DeepWalk | $\mathbf{A}$ | $68.5 \pm 0.5$ | $49.8 \pm 0.2$ | $66.2 \pm 0.7$ | $85.68 \pm 0.06$ | $89.44 \pm 0.11$ | $84.61 \pm 0.22$ |
| | GAE | $\mathbf{X, A}$ | $72.1 \pm 0.5$ | $66.5 \pm 0.4$ | $71.8 \pm 0.6$ | $85.27 \pm 0.19$ | $91.62 \pm 0.13$ | $90.01 \pm 0.71$ |
| | GMI | $\mathbf{X, A}$ | $83.0 \pm 0.3$ | $72.4 \pm 0.1$ | $79.9 \pm 0.2$ | $82.21 \pm 0.31$ | $90.68 \pm 0.17$ | OOM |
| | GRACE | $\mathbf{X, A}$ | $81.9 \pm 0.4$ | $71.3 \pm 0.3$ | $80.1 \pm 0.2$ | $86.53 \pm 0.28$ | $92.24 \pm 0.17$ | $92.98 \pm 0.05$ |
| | GCA | $\mathbf{X, A}$ | $81.7 \pm 0.3$ | $71.1 \pm 0.4$ | $79.5 \pm 0.5$ | $87.85 \pm 0.31$ | $92.49 \pm 0.09$ | $93.10 \pm 0.01$ |
| | GraphMAE | $\mathbf{X, A}$ | $84.2 \pm 0.4$ | $73.4 \pm 0.4$ | $81.1 \pm 0.4$ | $88.12 \pm 0.30$ | $92.97 \pm 0.21$ | $93.03 \pm 0.16$ |
| | G-BT | $\mathbf{X, A}$ | $84.0 \pm 0.4$ | $73.0 \pm 0.3$ | $80.7 \pm 0.4$ | $88.14 \pm 0.33$ | $92.63 \pm 0.44$ | $92.95 \pm 0.17$ |
| | CCA-SSG | $\mathbf{X, A}$ | $84.2 \pm 0.4$ | $73.1 \pm 0.3$ | $81.6 \pm 0.4$ | $88.74 \pm 0.28$ | $\mathbf{93.14 \pm 0.14}$ | $93.31 \pm 0.22$ |
| | InfoGCL | $\mathbf{X, A}$ | $83.5 \pm 0.3$ | $73.5 \pm 0.4$ | $79.1 \pm 0.2$ | - | - | - |
| | CorInfoMax | $\mathbf{X, A}$ | $82.6 \pm 0.4$ | $72.2 \pm 0.5$ | $80.4 \pm 0.4$ | $87.98 \pm 0.14$ | $92.63 \pm 0.10$ | $92.88 \pm 0.15$ |
| | MVGRL | $\mathbf{X, A}$ | $83.7 \pm 0.6$ | $73.6 \pm 0.3$ | $79.9 \pm 0.2$ | $87.52 \pm 0.11$ | $91.74 \pm 0.07$ | $92.11 \pm 0.12$ |
| | DGI | $\mathbf{X, A}$ | $82.3 \pm 0.6$ | $71.8 \pm 0.7$ | $76.8 \pm 0.6$ | $83.95 \pm 0.47$ | $91.61 \pm 0.22$ | $92.15 \pm 0.63$ |
| | GMIM | $\mathbf{X, A}$ | $83.4 \pm 0.6$ | $72.5 \pm 0.5$ | $81.6 \pm 0.5$ | $88.64 \pm 0.22$ | $92.95 \pm 0.17$ | $92.48 \pm 0.10$ |
| | GMIM-IC | $\mathbf{X, A}$ | $\mathbf{84.6 \pm 0.5}$ | $\mathbf{73.7 \pm 0.4}$ | $\mathbf{81.8 \pm 0.6}$ | $\mathbf{88.80 \pm 0.49}$ | $93.10 \pm 0.26$ | $\mathbf{93.45 \pm 0.17}$ |

## 6.2 COMPARISON EXPERIMENTS

Here, we compare our method with state-of-the-art baselines in terms of performance and efficiency.

**Performance Comparison.** To evaluate the effectiveness of our approach, we compare our method with the state-of-the-art baselines on node classification task under the simple linear classifier. The average classification accuracy with standard deviation of 20 results is reported for each dataset. We compare our approach with unsupervised methods including DeepWalk (Perozzi et al., 2014), GAE (Kipf & Welling, 2016b), DGI (Veličković et al., 2018), GMI (Peng et al., 2020), GRACE (Zhu et al., 2020), GCA (Zhu et al., 2021), G-BT (Bielak et al., 2022), CCA-SSG (Zhang et al., 2021) InfoGCL (Xu et al., 2021b), GraphMAE (Hou et al., 2022), CorInfoMax (Ozsoy et al., 2022) and MVGRL (Hassani & Khasahmadi, 2020). Furthermore, some supervised models including multi-layer perceptron (MLP), GCN (Kipf & Welling, 2016a), and GAT (Veličković et al., 2017) are also as baselines. We adopt the public splits on Cora, Citeseer and Pubmed, and a 1:1:8 split for training/validation/testing on the other three datasets. To make a fair comparison, for the methods without adopting the same splits as ours, we conduct experiments to get relevant results based on the officially released source code with a hyper-parameter search. Table 1 reports the classification results on six datasets. It can be observed that our method achieves high performance on all datasets and outperforms the state-of-the-art peers on five out of six datasets. These results clearly demonstrate the effectiveness of our approach. After subjecting the node representations to a rigorous statistical hypothesis testing, we discover that they do not actually conform to a Gaussian distribution (Refer to Appendix E for further details). In other words, our method remains highly effective in non-Gaussian scenarios. Overall, GMIM-IC surpasses GMIM. One reason is that the identity constraint imposes stricter demands on cross-view consistency and aligns with the practical design of the shared network architecture. Besides, GMIM-IC demonstrates comparable performance with CCA-SSG, which can serve as empirical support for our theoretical analysis.

**Efficiency Comparison.** Please refer to Appendix H.1.

## 6.3 HYPERPARAMETER SENSITIVITY ANALYSIS AND EXPLORATORY EXPERIMENTS

**Effect of representation dimension.** We conduct experiments by varying the representation dimension to investigate its impacts on performance. Figure 2 summarizes the results of the three variants based on Eq. (2), Eq. (3), and Eq. (5) on four datasets. It can be observed that our method achieves optimal performance with an appropriately large dimension, because the representations exhibit better discriminability and linear separability in high-dimensional space. However, as the dimension becomes excessively large such as 1,024, there is a slight decrease in performance. This can be blamed on the fact that an excessively high representation dimension hinders the model from learning compact and information-dense representations. Another non-negligible underlying factor

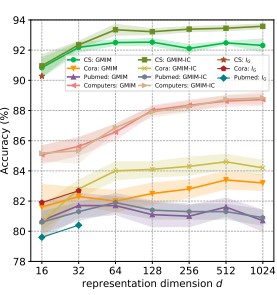
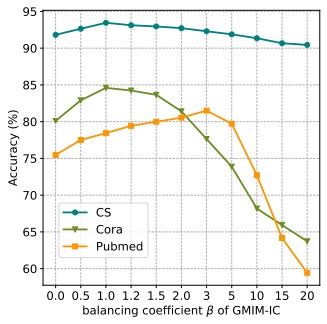
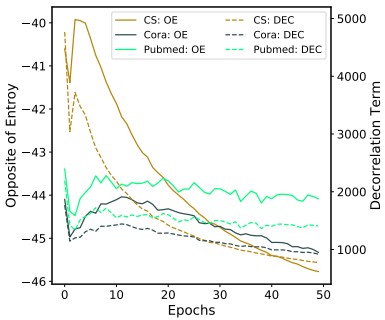

Figure 2: Effect of representation dimension. "$I_G$" denotes the results based on Eq. (2).

Figure 3: The classification accuracy of GMIM-IC under varying balancing coefficients.

Figure 4: Synergistic changes between opposite of entropy (OE) and decorrelation loss (DEC).

for declining performance is that higher dimensions lead to poorer estimation of the covariance matrix. Even in low-dimensional settings, our method still delivers decent performance. This finding can be attributed to the effective maximization of information entropy, which prevents dimensional collapse, enhances the diversity of representations, and ultimately improve model performance within limited dimensions. Under the objective based on Eq. (2), the results in high-dimensional settings and on Computers are unavailable. In such scenarios, the covariance matrix exhibits numerous small eigenvalues, causing its determinant to approach zero. This phenomenon introduces numerical instability and eventually disrupts training process.

**Impact of balancing coefficient.** We study the impacts of the balancing coefficient $\beta$ in $\mathcal{L}_{\texttt{GMIM-IC}}$ on performance. Figure 3 illustrates the variation of classification accuracy with varying values of the coefficient. The performance exhibits a pattern of initially increasing and later decreasing as $\beta$ goes up. When $\beta$ is small, the entropy maximization term cannot fully exploit its role in promoting diversity of representations. When $\beta$ is too large, too much emphasis on maximizing information entropy leads to informative yet meaningless representations.

**Synergistic changes between opposite of entropy and decorrelation loss.** Taking $\mathcal{L}_{\texttt{GMIM-IC}}$ as the optimization objective, we visualize the joint changes of decorrelation loss in Eq. (6) and opposite of entropy in Eq. (5). For each dataset in Figure 4, the decorrelation loss (dashed line) exhibits a nearly identical trend to the opposite of entropy. Experimental observations potentially indicate a similar effect between them, which can serve as an empirical support for Section 4.

## 7 LIMITATIONS, CONCLUSION, AND FUTURE WORK

**Limitations.** Due to extreme limitations in computational resources, we only conducted empirical studies on graphs. Extension experiments on other types of data, such as images, are left for future.

**Conclusion.** In this paper, we have presented a graph contrastive learning method under the common Gaussian assumption for node representations, which does not rely on any parametric mutual information estimators and negative samples. Furthermore, we provide two theoretical explanations regarding the relationship between decorrelation-based methods and contrastive-based methods. Our analysis reveals that the decorrelation-based method can be interpreted as a variant of contrastive methods when the Gaussian assumption and identity constraint are considered. Extensive comparative experiments and visual analysis have demonstrated the effectiveness, efficiency, and theoretical soundness of our method. Overall, the Gaussian assumption motivates our research, but empirical evidence demonstrates the continued effectiveness of our method in non-Gaussian scenarios, which extends the practical application scope of our work.

**Future Work.** Our research paves a new path for graph self-supervised learning. The prospect of extending the Gaussian assumption to other distributions, such as the Gamma distribution, stands as a viable endeavor. Furthermore, the exploration of relationships among distinct variants under different distributions represents a valuable and exciting pursuit. In addition, enhancing the reliability of covariance matrix estimation (Ledoit & Wolf, 2020) is a promising aspect for improving our method.

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

## A    DERIVATIONS OF GAUSSIAN MUTUAL INFORMATION

The formal derivation relies on the following lemma:

**Lemma 2.** *For matrices* $\mathbf{A} \in \mathbb{R}^{N \times K}$ *and* $\mathbf{B} \in \mathbb{R}^{K \times N}$,

$$\mathrm{tr}(\mathbf{AB}) = \mathrm{tr}(\mathbf{BA}), \tag{11}$$

*where* $\mathrm{tr}(\cdot)$ *denotes the trace of a matrix.*

*Proof.*

$$\mathrm{tr}(\mathbf{AB}) = \sum_{i=1}^{N} \sum_{j=1}^{K} A_{ij} \cdot B_{ji} = \sum_{j=1}^{K} \sum_{i=1}^{N} B_{ji} \cdot A_{ij} = \mathrm{tr}(\mathbf{BA}).$$

$\square$

The mutual information $I_G(X; Y)$ can be expanded as follows:

$$
\begin{aligned}
I_G(X; Y) &= \int_{\mathcal{X}} \int_{\mathcal{Y}} p_{x,y}(X, Y) \ln \frac{p_{x,y}(X, Y)}{p_x(X) \cdot p_y(Y)} dX dY \\
&= \int_{\mathcal{X}} \int_{\mathcal{Y}} p_{x,y}(X, Y) \ln p_{x,y}(X, Y) dX dY \\
&\quad - \int_{\mathcal{X}} \int_{\mathcal{Y}} p_{x,y}(X, Y) \ln p_x(X) dX dY \\
&\quad - \int_{\mathcal{X}} \int_{\mathcal{Y}} p_{x,y}(X, Y) \ln p_y(Y) dX dY.
\end{aligned}
\tag{12}
$$

In order to obtain the desired result, we will perform integration on the three terms in Eq. (12), respectively.

For a Gaussian variable $X$ with mean $\boldsymbol{\mu}_X$ and covariance matrix $\boldsymbol{\Sigma}_X$, its probability density function can be expressed as

$$p_x(X) = \frac{1}{\sqrt{(2\pi)^d \det(\boldsymbol{\Sigma}_X)}} \exp\left(-\frac{1}{2}(X - \boldsymbol{\mu}_X)^T \boldsymbol{\Sigma}_X^{-1}(X - \boldsymbol{\mu}_X)\right). \tag{13}$$

Thus, it can be known that

$$
\begin{aligned}
&- \int_{\mathcal{X}} \int_{\mathcal{Y}} p_{x,y}(X, Y) \ln p_x(X) dX dY \\
={}& - \int_{\mathcal{X}} \int_{\mathcal{Y}} p_{x,y}(X, Y) dY \ln p_x(X) dX \\
={}& - \int_{\mathcal{X}} p_x(X) \ln p_x(X) dX \\
={}& - \int_{\mathcal{X}} p_x(X) \left( \ln \frac{1}{\sqrt{(2\pi)^d \det(\boldsymbol{\Sigma}_X)}} - \frac{1}{2}(X - \boldsymbol{\mu}_X)^T \boldsymbol{\Sigma}_X^{-1}(X - \boldsymbol{\mu}_X) \right) dX \\
={}& \frac{1}{2} \int_{\mathcal{X}} p_x(X) \ln\left((2\pi)^d \det(\boldsymbol{\Sigma}_X)\right) dX + \frac{1}{2} \int_{\mathcal{X}} p_x(X)(X - \boldsymbol{\mu}_X)^T \boldsymbol{\Sigma}_X^{-1}(X - \boldsymbol{\mu}_X) dX \\
={}& \frac{\ln\left((2\pi)^d \det(\boldsymbol{\Sigma}_X)\right)}{2} \int_{\mathcal{X}} p_x(X) dX + \frac{1}{2} \int_{\mathcal{X}} p_x(X)(X - \boldsymbol{\mu}_X)^T \boldsymbol{\Sigma}_X^{-1}(X - \boldsymbol{\mu}_X) dX \\
={}& \frac{1}{2} \ln \det(\boldsymbol{\Sigma}_X) + \frac{d}{2} \ln(2\pi) + \frac{1}{2} \int_{\mathcal{X}} p_x(X)(X - \boldsymbol{\mu}_X)^T \boldsymbol{\Sigma}_X^{-1}(X - \boldsymbol{\mu}_X) dX.
\end{aligned}
\tag{14}
$$

We will deal specially with $\int_{\mathcal{X}} p_x(X)(X - \boldsymbol{\mu}_X)^T \boldsymbol{\Sigma}_X^{-1}(X - \boldsymbol{\mu}_X) dX$. Actually, $X - \boldsymbol{\mu}_X$ is vector $\in \mathbb{R}^d$ and $(X - \boldsymbol{\mu}_X)^T \boldsymbol{\Sigma}_X^{-1}(X - \boldsymbol{\mu}_X)$ is a scalar value. If we regard $X - \boldsymbol{\mu}_X$ as a matrix $\in \mathbb{R}^{d \times 1}$,

$(X - \boldsymbol{\mu}_X)^T \boldsymbol{\Sigma}_X^{-1}(X - \boldsymbol{\mu}_X)$ will be a matrix $\in \mathbb{R}^{1 \times 1}$. The original expression can be rephrased as $tr\big((X - \boldsymbol{\mu}_X)^T \boldsymbol{\Sigma}_X^{-1}(X - \boldsymbol{\mu}_X)\big)$. Taking $(X - \boldsymbol{\mu}_X)^T$ as $\mathbf{A}$ in Eq. (11) and $\boldsymbol{\Sigma}_X^{-1}(X - \boldsymbol{\mu}_X)$ as $\mathbf{B}$, respectively, we can know that

$$
\begin{aligned}
&\int_{\mathcal{X}} p_x(X) tr\Big((X - \boldsymbol{\mu}_X)^T \boldsymbol{\Sigma}_X^{-1}(X - \boldsymbol{\mu}_X)\Big) dX \\
&= \int_{\mathcal{X}} p_x(X) tr\Big(\boldsymbol{\Sigma}_X^{-1}(X - \boldsymbol{\mu}_X)(X - \boldsymbol{\mu}_X)^T\Big) dX \\
&= tr\Big(\boldsymbol{\Sigma}_X^{-1} \int_{\mathcal{X}} p_x(X)(X - \boldsymbol{\mu}_X)(X - \boldsymbol{\mu}_X)^T dX\Big) \\
&= tr\Big(\boldsymbol{\Sigma}_X^{-1} \boldsymbol{\Sigma}_X\Big) \\
&= d
\end{aligned}
\tag{15}
$$

Plugging the result of Eq. (15) into Eq. (14), it can be concluded that

$$
-\int_{\mathcal{X}}\int_{\mathcal{Y}} p_{x,y}(X,Y) \ln p_x(X) dX dY = \frac{1}{2} \ln \det(\boldsymbol{\Sigma}_X) + \frac{d}{2}\ln(2\pi) + \frac{d}{2} = \frac{1}{2}\ln\det(\boldsymbol{\Sigma}_X) + \frac{d}{2}\ln(2\pi e).
\tag{16}
$$

Symmetrically, it can be obtained that

$$
-\int_{\mathcal{X}}\int_{\mathcal{Y}} p_{x,y}(X,Y) \ln p_y(Y) dX dY = \frac{1}{2}\ln\det(\boldsymbol{\Sigma}_Y) + \frac{d}{2}\ln(2\pi e).
\tag{17}
$$

Similarly,

$$
\int_{\mathcal{X}}\int_{\mathcal{Y}} p_{x,y}(X,Y) \ln p_{x,y}(X,Y) dX dY = -\frac{1}{2}\ln\det(\boldsymbol{\Sigma}_{X,Y}) - \frac{d+d}{2}\ln(2\pi e).
\tag{18}
$$

Plugging Eq. (16), (17), and (18) into $I_G(X;Y)$ in Eq. (12), it results in the following closed-form of Gaussian mutual information:

$$
I_G(X;Y) = \frac{1}{2}\ln\frac{\det(\boldsymbol{\Sigma}_X) \cdot \det(\boldsymbol{\Sigma}_Y)}{\det(\boldsymbol{\Sigma}_{X,Y})}.
\tag{19}
$$

The derivation of Gaussian mutual information can also be referenced from other literature (Bouhlel & Dziri, 2019; Zhang et al., 2023; Zhouyin & Liu, 2021; Cover, 1999). However, few studies have provided a complete derivation process. Thus, for the self-completeness of this paper, we give the complete derivation here.

## B  DERIVATION AND ANALYSIS ABOUT GMIM-IC

According to Property 1, the Gaussian mutual information $I_G(X;Y) = \frac{1}{2}\ln\frac{\det(\boldsymbol{\Sigma}_X)\cdot\det(\boldsymbol{\Sigma}_Y)}{\det(\boldsymbol{\Sigma}_{X,Y})}$ can be restated as

$$
I_G(X;Y) = H_G(X) - H_G(X|Y),
\tag{20}
$$

where $H_G(X) = -\int_{\mathcal{X}} p_x(X)\ln p_x(X) dX$ is the entropy of $X$ and $H_G(X|Y)$ is its conditional entropy given $Y$. Based on Eq. (14) and (16), $H_G(X) = \frac{1}{2}\ln\det(\boldsymbol{\Sigma}_X) + \frac{d}{2}\ln(2\pi e)$, that is, $H_G(X) \propto \ln\det(\boldsymbol{\Sigma}_X)$. As mentioned in the main text, directly optimizing $\ln\det(\boldsymbol{\Sigma}_X)$ can lead to numerical instability. After adjusting the eigenvalues by applying shifting and scaling operations, we can obtain a feasible substitution $\ln\det(\mathbf{I} + \eta \cdot \boldsymbol{\Sigma}_X)$. Therefore, maximizing $\ln\det(\mathbf{I} + \eta \cdot \boldsymbol{\Sigma}_X)$ can be equivalent to increasing the entropy $H_G(X)$.

As discussed in the main text, the conditional entropy $H_G(X|Y)$ is minimized when the relationship between $X$ and $Y$ can be determined by a function. Considering the prior of network design, which has two shared branches, we expect that this function is an identity mapping. Concretely, this is realized by imposing identity constraint to Eq. (20). In our practice, the node representations from view $A$ can be regarded as $N$ empirical samples of $X$ while those from view $B$ are related to $Y$.

Taking all the above factors into consideration, we can derive an objective based on Eq. (20):

$$\mathcal{L}_{\texttt{GMIM-IC}}^{\texttt{A}} = \frac{1}{N} \sum_{v \in \mathcal{V}} \|\mathbf{h}_v^A - \mathbf{h}_v^B\|_2^2 - \gamma \cdot \ln \det(\mathbf{I} + \eta \cdot \boldsymbol{\Sigma}_A), \tag{21}$$

where $\gamma$ indicates a balancing factor. Symmetrically, we can obtain an objective $\mathcal{L}_{\texttt{GMIM-IC}}^{\texttt{B}}$ corresponding to view $B$. Combining the two terms, it results in

$$\mathcal{L}_{\texttt{GMIM-IC}} = \frac{1}{N} \sum_{v \in \mathcal{V}} \|\mathbf{h}_v^A - \mathbf{h}_v^B\|_2^2 - \beta \cdot \left( \ln \det(\mathbf{I} + \eta \cdot \boldsymbol{\Sigma}_A) + \ln \det(\mathbf{I} + \eta \cdot \boldsymbol{\Sigma}_B) \right). \tag{22}$$

Minimizing the objective $\mathcal{L}_{\texttt{GMIM-IC}}$ is equivalent to maximizing Gaussian mutual information while imposing identity constraint across various views.

## C  Proofs and Derivations in Section 4 and 5

### C.1  Proof of Lemma 1

For convenience, we restate Lemma 1:

**Lemma 1.** *For a square matrix* $\mathbf{M}$*,* $\det(\exp(\mathbf{M})) = \exp(\text{tr}(\mathbf{M}))$*. Replace* $\mathbf{M}$ *with* $\ln(\mathbf{I} + \eta \cdot \boldsymbol{\Sigma}_*)$ *:*

$$\ln \det(\mathbf{I} + \eta \cdot \boldsymbol{\Sigma}_*) = \text{tr}(\ln(\mathbf{I} + \eta \cdot \boldsymbol{\Sigma}_*)), \tag{23}$$

*where* $* \in \{A, B\}$*. Applying Taylor expression to the logarithmic function in* $\text{tr}(\ln(\mathbf{I} + \eta \cdot \boldsymbol{\Sigma}_*))$*, it can be known that*

$$\ln \det(\mathbf{I} + \eta \cdot \boldsymbol{\Sigma}_*) = tr \left( \sum_{k=1}^{+\infty} \frac{(-1)^{k+1}}{k} (\eta \cdot \boldsymbol{\Sigma}_*)^k \right). \tag{24}$$

*Proof.* Assuming $\{\lambda_1', \lambda_2', \ldots, \lambda_d'\}$ are $d$ eigenvalues of the matrix $\mathbf{M}$, $\{e^{\lambda_1'}, e^{\lambda_2'}, \ldots, e^{\lambda_d'}\}$ are $d$ eigenvalues of the matrix $\exp(\mathbf{M})$ accordingly. Thus, $\det(\exp(\mathbf{M})) = \prod_{i=1}^{d} e^{\lambda_i'} = \exp(\sum_{i=1}^{d} \lambda_i') = \exp(\text{tr}(\mathbf{M}))$. Taking $\mathbf{M} = \ln(\mathbf{I} + \eta \cdot \boldsymbol{\Sigma}_*)$, we can obtain $\det(\mathbf{I} + \eta \cdot \boldsymbol{\Sigma}_*) = \exp(\text{tr}(\ln(\mathbf{I} + \eta \cdot \boldsymbol{\Sigma}_*)))$, that is, $\ln \det(\mathbf{I} + \eta \cdot \boldsymbol{\Sigma}_*) = \text{tr}(\ln(\mathbf{I} + \eta \cdot \boldsymbol{\Sigma}_*))$.

Applying the Taylor expression $\ln(1 + x) = \sum_{k=1}^{\infty} \frac{(-1)^{k+1} \cdot x^k}{k}$, we have

$$\begin{aligned} &\ln \det(\mathbf{I} + \eta \cdot \boldsymbol{\Sigma}_*) \\ =& \text{tr}(\ln(\mathbf{I} + \eta \cdot \boldsymbol{\Sigma}_*)) \\ =& tr \left( \sum_{k=1}^{+\infty} \frac{(-1)^{k+1}}{k} (\eta \cdot \boldsymbol{\Sigma}_*)^k \right). \end{aligned} \tag{25}$$

$\square$

Furthermore, we can obtain a second-order Taylor approximation:

$$\begin{aligned} &- \ln \det(\mathbf{I} + \eta \cdot \boldsymbol{\Sigma}_*) \\ \approx& - tr \left( \sum_{k=1}^{2} \frac{(-1)^{k+1}}{k} (\eta \cdot \boldsymbol{\Sigma}_*)^k \right) \\ =& \frac{\eta^2}{2} \cdot tr \left( (\boldsymbol{\Sigma}_*)^2 \right) - \eta \cdot tr(\boldsymbol{\Sigma}_*) \\ =& \frac{\eta^2}{2} \cdot \|\boldsymbol{\Sigma}_*\|_F^2 - \eta \cdot d \\ =& \frac{\eta^2}{2} \cdot \sum_{i=1}^{d} \sum_{j=1, j \neq i}^{d} (\Sigma_*^{ij})^2 + \frac{\eta^2}{2} \cdot d - \eta \cdot d. \end{aligned} \tag{26}$$

Ignoring constant terms, $\sum_{i=1}^{d} \sum_{j=1, j \neq i}^{d} (\Sigma_*^{ij})^2$ is equivalent to a second-order Taylor expansion of $-\ln \det(\mathbf{I} + \eta \cdot \mathbf{\Sigma}_*)$. Thus, minimizing $\sum_{i=1}^{d} \sum_{j=1, j \neq i}^{d} (\Sigma_*^{ij})^2$ has a similar effect to reducing $-\ln \det(\mathbf{I} + \eta \cdot \mathbf{\Sigma}_*)$.

We have completed the entire deviation.

## C.2 PROOF OF PROPOSITION 2

The formal proof of Proposition 2 relies on the following lemma:

**Lemma 3.** *For a real symmetric matrix $\mathbf{A}$ whose eigenvalues are all 1, it must be the identity matrix.*

*Proof.* For a real symmetric matrix $\mathbf{A}$, it can be diagonalized by an orthogonal matrix, that is, $\mathbf{A} = \mathbf{U}\mathbf{D}\mathbf{U}^\top$ with the orthogonal matrix $\mathbf{U}$ and the diagonal matrix $\mathbf{D}$. Since the eigenvalues of $\mathbf{A}$ are all 1, $\mathbf{D}$ is equal to an identity matrix $\mathbf{I}$. Thus, $\mathbf{A} = \mathbf{U}\mathbf{I}\mathbf{U}^\top = \mathbf{I}$. □

For convenience, we restate Proposition 2 here.

**Proposition 2.** *When $\ln \det(\mathbf{I} + \eta \cdot \mathbf{\Sigma}_*)$ or $\ln \det(\mathbf{\Sigma}_*)$ is maximized, the empirical covariance matrix $\mathbf{\Sigma}_*$ will converge to an identity matrix.*

*Proof.* Assuming $\{\lambda_1, \lambda_2, \ldots, \lambda_d\}$ are $d$ eigenvalues of the covariance matrix $\mathbf{\Sigma}_*$, $\det(\mathbf{I} + \eta \cdot \mathbf{\Sigma}_*) = \prod_{i=1}^{d}(1 + \eta \cdot \lambda_i)$. Besides, $\sum_{i=1}^{d}(1 + \eta \cdot \lambda_i) = \text{tr}(\mathbf{I} + \eta \cdot \mathbf{\Sigma}_*) = d + \eta \cdot d$. According to the AM-GM Inequality (Hirschhorn, 2007), it can be known that

$$
\begin{aligned}
&\det(\mathbf{I} + \eta \cdot \mathbf{\Sigma}_*) \\
&= \prod_{i=1}^{d}(1 + \eta \cdot \lambda_i) \\
&\leq \left( \frac{1 + \eta \cdot \lambda_1 + 1 + \eta \cdot \lambda_2 + \cdots + 1 + \eta \cdot \lambda_d}{d} \right)^d \\
&= (1 + \eta)^d.
\end{aligned}
\tag{27}
$$

$\det(\mathbf{I} + \eta \cdot \mathbf{\Sigma}_*)$ achieves the upper bound of $(1 + \eta)^d$ when the eigenvalues $\{\lambda_1, \ldots, \lambda_d\}$ of $\mathbf{\Sigma}_*$ are all equal to 1. Similarly, applying the above derivation to $\det(\mathbf{\Sigma}_*)$, we can easily conclude that $\det(\mathbf{\Sigma}_*)$ reaches a maximum value of 1 when all eigenvalues are equal to 1.

$\mathbf{\Sigma}_* = \frac{1}{N} \mathbf{H}_*^\top \mathbf{H}_*$ is a real symmetric matrix. According to Lemma 3, $\mathbf{\Sigma}_*$ will converge to the identity matrix when its eigenvalues are all equal to 1. Thus, we conclude the proof. □

## C.3 PROOF OF PROPERTY 2 AND FURTHER STATEMENT

**Property 2.** *For empirical covariance matrix $\mathbf{\Sigma} = \frac{1}{N} \mathbf{H}^\top \mathbf{H} \in \mathbb{R}^{d \times d}$ with batch-normalized representations $\mathbf{H} = [\mathbf{h}_1, ..., \mathbf{h}_N]^\top \in \mathbb{R}^{N \times d}$, which has $d$ eigenvalues $[\lambda_1, \lambda_2, \ldots, \lambda_d]$ corresponding to $d$ eigenvectors $[\mathbf{q}_1, \mathbf{q}_2, \ldots, \mathbf{q}_d]$, the variance of data $\mathbf{H}$ along the $k$-th principal direction (that is, the direction of $\mathbf{q}_k$) is numerically equal to $\lambda_k$.*

*Proof.* For $N$ $d$-dimensional data points $\mathbf{H} = [\mathbf{h}_1, \ldots, \mathbf{h}_N]^\top \in \mathbb{R}^{N \times d}$, which has been normalized to 0-mean and 1-standard-deviation along sample direction (*i.e.*, $\frac{1}{N} \sum_{i=1}^{N} \mathbf{h}_i = \mathbf{0}$), its covariance matrix is $\mathbf{\Sigma} = \frac{1}{N} \mathbf{H}^\top \mathbf{H}$. After eigendecomposition for $\mathbf{\Sigma}$, we can obtain $d$ unit orthogonal eigenvectors $[\mathbf{q}_1, \ldots, \mathbf{q}_d]$ associated to eigenvalues $[\lambda_1, \ldots, \lambda_d]$, respectively. According to $\frac{1}{N} \mathbf{H}^\top \mathbf{H} \mathbf{q}_k = \lambda_k \mathbf{q}_k$, it can be known that

$$
\frac{1}{N} \mathbf{q}_k^\top \mathbf{H}^\top \mathbf{H} \mathbf{q}_k = \lambda_k \mathbf{q}_k^\top \mathbf{q}_k = \lambda_k.
\tag{28}
$$

Taking a principal direction $\mathbf{q}_k$ as explanation, the projection of a sample $\mathbf{h}_i$ onto this direction is $z_i = \mathbf{q}_k^\top \mathbf{h}_i$, and the mean of all projections is

$$
\bar{z} = \frac{1}{N} \sum_{i=1}^{N} z_i = \frac{1}{N} \sum_{i=1}^{N} \mathbf{q}_k^\top \mathbf{h}_i = 0.
\tag{29}
$$

Thus, along the principal direction $\mathbf{q}_k$, the variance is

$$
\begin{aligned}
\frac{1}{N}\sum_{i=1}^{N}(z_i - \bar{z})^2 \\
=&\frac{1}{N}\sum_{i=1}^{N}\mathbf{q}_k^\top \mathbf{h}_i \mathbf{h}_i^\top \mathbf{q}_k \\
=&\frac{1}{N}\mathbf{q}_k^\top (\sum_{i=1}^{N}\mathbf{h}_i \mathbf{h}_i^\top)\mathbf{q}_k \\
=&\frac{1}{N}\mathbf{q}_k^\top \mathbf{H}^\top \mathbf{H}\mathbf{q}_k \\
=&\lambda_k.
\end{aligned}
\tag{30}
$$

The above equation demonstrates that the variance of data $\mathbf{H}$ along the direction $\mathbf{q}_k$ is equal to $\lambda_k$. Thus, the proof is concluded. □

### C.4 Proof of Proposition 3

For convenience, we restate Proposition 3 here.

**Proposition 3.** *When the representations scatter over the unit hypersphere $\mathcal{S}^{d-1}$ uniformly (that is, they obey a complete uniform distribution), their entropy will reach the maximum value.*

*Proof.* Assuming that the representations follow a distribution $p(X)$ on the unit hypersphere $\mathcal{S}^{d-1}$, proving Proposition 3 is equivalent to demonstrate that when $p(X)$ is a uniform distribution, the entropy of variable $X$ is maximized. The corresponding mathematical expression can be stated as follows:

$$
\begin{aligned}
\max \quad & -\int_{\mathcal{S}^{d-1}} p(X)\ln p(X)dX \\
s.t. \quad & \int_{\mathcal{S}^{d-1}} p(X)dX = 1
\end{aligned}
\tag{31}
$$

To find the optimal form of $p(X)$ subject to the constraint $\int_{\mathcal{S}^{d-1}} p(X)dX = 1$, we construct the following Lagrangian function:

$$
L(p(X),\lambda) = -\int_{\mathcal{S}^{d-1}} p(X)\ln p(X)dX + \lambda \cdot \left(\int_{\mathcal{S}^{d-1}} p(X)dX - 1\right),
\tag{32}
$$

where $\lambda$ denotes Lagrange multiplier.

Taking the derivative of the Lagrangian function $L(p(X),\lambda)$ with respect to $p(X)$ and setting it equal to zero, we know that

$$
\frac{\partial L(p(X),\lambda)}{\partial p(X)} = -\ln p(X) - 1 + \lambda = 0.
\tag{33}
$$

Hence, the optimal form of the probability density function is

$$
p(X) = e^{\lambda - 1}.
\tag{34}
$$

To satisfy the constraint $\int_{\mathcal{S}^{d-1}} p(X)dX = 1$, we have

$$
\int_{\mathcal{S}^{d-1}} p(X)dX = \int_{\mathcal{S}^{d-1}} e^{\lambda - 1}dX = e^{\lambda - 1}\int_{\mathcal{S}^{d-1}} dX = 1.
\tag{35}
$$

Letting $S = \int_{\mathcal{S}^{d-1}} dX$ represent the surface area of the unit hypersphere $\mathcal{S}^{d-1}$, we can know that

$$
\lambda = 1 - \ln S.
\tag{36}
$$

According to (Rennie, 2005), we can obtain that $S = \frac{2\pi^{d/2}}{\Gamma(d/2)}$, where $\Gamma(\cdot)$ denotes the gamma function. Taking Eq. (36) into Eq. (34), it can be known that

$$
p(X) = \frac{1}{S} = \frac{\Gamma(d/2)}{2\pi^{d/2}},
\tag{37}
$$

which is a uniform distribution on the unit hypersphere. Thus, it can be known that the entropy of representations on the unit hypersphere reach the maximum value when they obey a uniform distribution. We conclude the proof. □

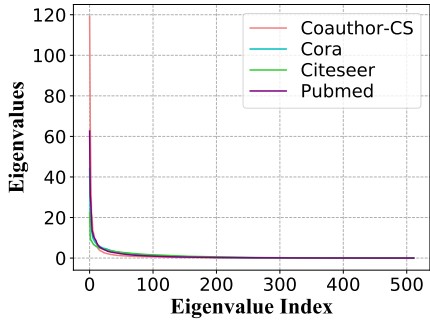

Figure 5: Eigenvalues of covariance matrices of node representations from the randomly initial representation encoder.

---

**Algorithm 1** Hypothesis Testing based on scipy.

---

```
import numpy as np
from scipy import stats

# H: node representation matrix with the size of (N, d)
ret = stats.normaltest(H, axis=0)[1] # results with the shape of (d,)
```

---

## D   VISUALIZATIONS OF INITIAL EIGENVALUES

In the main text, we have pointed out that it is inappropriate to directly construct an objective function (that is, $\mathcal{L}_G = \ln \frac{\det(\mathbf{\Sigma}_{A,B})}{\det(\mathbf{\Sigma}_A) \cdot \det(\mathbf{\Sigma}_B)}$) based on $I_G(X;Y) = \frac{1}{2} \ln \frac{\det(\mathbf{\Sigma}_X) \cdot \det(\mathbf{\Sigma}_Y)}{\det(\mathbf{\Sigma}_{X,Y})}$, as it would result in numerical instability.

Figure 5 visualizes the eigenvalues of covariance matrices of node representations in the initial epoch of the pretraining phase. It can be observed that a significant portion of the eigenvalues are close to or equal to zero. This phenomenon leads to the determinant of the covariance matrix, which is numerically equivalent to the product of all eigenvalues, being zero. Therefore, including the determinant of the original covariance matrix in the objective function will potentially lead to computational instability. Our empirical experiments suggest that when representation dimension is greater than $64$, the determinant of the original covariance matrix becomes zero for nearly all datasets.

## E   DISCUSSION ABOUT THE GAUSSIAN ASSUMPTION

In a previous research, we conducted a visualization of the histograms of representations, as illustrated in Figure 6. At first glance, the distribution of representations exhibits a Gaussian appearance. This observation sparked our curiosity about the possibility of directly calculating mutual information between two variables (*i.e.*, two views) under the Gaussian assumption. In general circumstances, mutual information cannot be directly computed and the current contrastive learning methods rely on additional neural estimators to approximate a lower bound. If our idea proves effective, it will substantially simplify existing methods. Without disappointment, extensive empirical studies demonstrate the effectiveness of our approach. Subsequently, we conducted a rigorous hypothesis testing on individual channels of representation matrices of multiple datasets based on library `scipy`, as shown in Algorithm 1. The outcomes indicated that node representations do not actually conform to a Gaussian distribution. This result is, in fact, promising, which implies that our approach will no longer be confined to Gaussian scenarios. In summary, visualized histograms and the Gaussian assumption provided the initial impetus for our research, while the fact that our approach still remains its performance under the non-Gaussian conditions extends the application scenarios of our method.

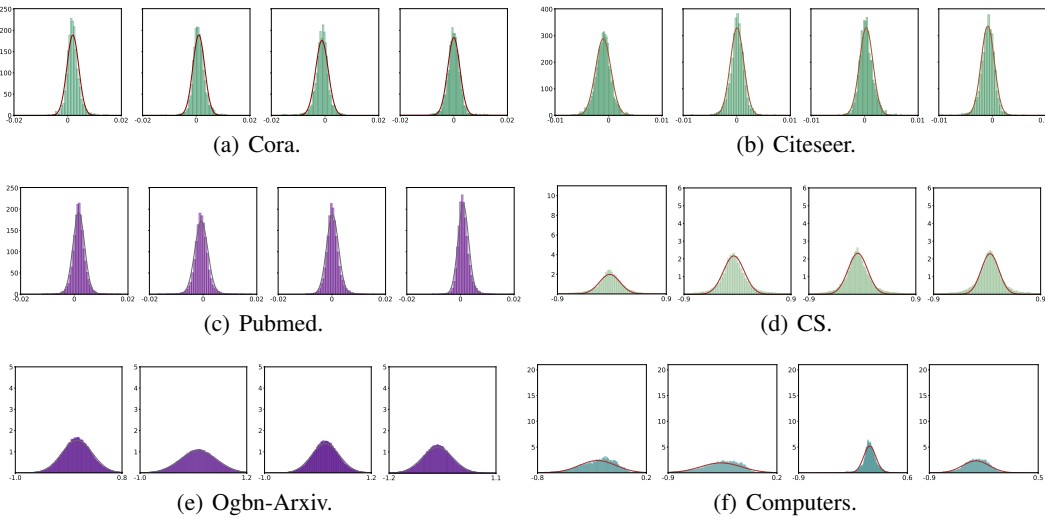

Figure 6: Histograms of individual representation channels on six datasets. Each curve represents a Gaussian distribution with mean and variance from the corresponding histogram. The histograms appear to exhibit a Gaussian appearance, but based on rigorous statistical tests, they do not strictly adhere to Gaussian distributions.

## F  COMPARISON WITH A PEER WORK AND RELATION WITH DISENTANGLED REPRESENTATION LEARNING

### F.1  COMPARISON WITH A PEER WORK

Towards the conclusion of our work, we observed that a peer study (Ozsoy et al., 2022), called CorInfoMax, shares certain similarities with our method, especially in terms of the objective function. Despite the similarities, the two works still exhibit significant distinctions as follows:

**Motivation.**  As described in the main text and Appendix E, our research starts from the Gaussian assumption with the expectation of enabling direct computation of mutual information, thereby eliminating mutual information neural estimators and simplifying existing contrastive learning methods. The core motivation of CorInfoMax is to leverage information entropy maximization to overcome the collapse issues and learn informative representations, as can be reflected in the title, abstract, and introduction of their paper.

**Network Architecture.**  Our method does not utilize additional projection heads. Our loss function directly operates on the output representations of the encoder $f_\theta(\cdot)$, while CorInfoMax first employs a projector of 3-layer MLP to map the outputs of the encoder to a new embedding space and then calculate loss function in the new space. Our approach reduces model complexity and enhances efficiency by directly optimizing the output space of the encoder. This distinction is not only reflected in the variations in network architecture but, more importantly, it actually indicates the disparities in motivations and underlying concepts between the two works. The purpose of our research is to enable the direct calculation of mutual information under the Gaussian assumption without relying on estimators and extra designs. The projectors introduced in CorInfoMax completely deviate from our initial motivation.

**Numerical Stability During Training.**  An important issue we addressed in our research is that directly incorporating the logarithm of the determinant of the covariance matrix in the objective function can lead to numerical instability, which is detailedly analyzed and discussed in Subsection 3.2 and Appendix D. To cope with this issue, we adopt the strategy of *offsetting and scaling eigenvalues* to be around 1. CorInfoMax turns to *adding a disturbance* in their objective. The comparisons between the two strategies are placed in Figure 7. When dimensions are higher than 128, the training

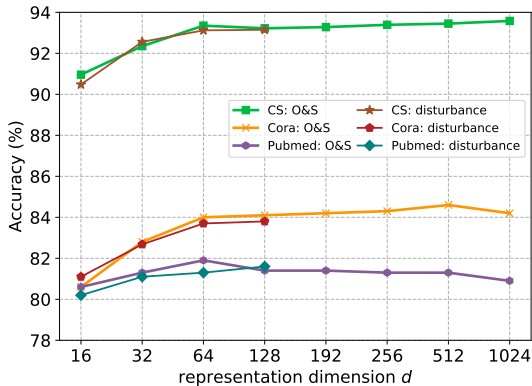

Figure 7: Performance of two strategies under various dimensions. $O\&S$: offsetting and scaling eigenvalues in our work; disturbance: the strategy of adding a disturbance in CorInfoMax. For "disturbance", when dimension is set to higher than 128, the training process is terminated due to numerical instability, and the results are not obtainable.

process under the strategy of adding disturbance is terminated due to numerical instability, because it cannot change the fact that many eigenvalues remain very close to 0.

**Explanations of Relationship Between Various SSL Methods.** We establish connections between contrastive and decorrelation-based methods, providing an explanation for decorrelation-based methods from the perspective of MI maximization. These contributions are of significant importance in establishing a unified theoretical framework for self-supervised learning methods. The work (Ozsoy et al., 2022) does not involve these aspects.

**Research on Dimensional Collapse.** In Section 5.1, we theoretically demonstrate from the perspective of eigen spectrum that the entropy maximization term can prevent dimensional collapse. In Section 5.2, we show that *InfoNCE* potentially maximizes information entropy. The work (Ozsoy et al., 2022) does not involve these contents, which primarily provides empirical evidence for the effectiveness of their method in preventing dimension collapse.

Finally, we express our gratitude to the authors of CorInfoMax for their outstanding contributions to the self-supervised learning community.

### F.2 RELATION WITH DISENTANGLED REPRESENTATION LEARNING

Our method expects to enforce the representations to achieve an isotropic Gaussian distribution with the aim of decoupling various dimensions and learning diverse representations, which is closely linked to a branch of deep learning called Disentangled Representation Learning (DRL) (Reddy et al., 2022; Shen et al., 2022; Bengio et al., 2013). The objective of disentangled representation learning is to achieve a clear separation of the distinct, independent, and informative generative factors inherent in the data (Bengio et al., 2013). DRL emphasizes the statistical independence among latent variables, which can be traced back to Independent Component Analysis (ICA) (Stone, 2004).

Independent Component Analysis, a computationally efficient Blind Source Separation (BSS) (Naik et al., 2014) technique, thinks that the observed mixed signals are obtained through a linear combination of source signals and aims to recover latent variables from observations. The traditional ICA assumes that the source signals follow non-Gaussian distributions and are statistically independent. The assumption of statistical independence among latent variables is, in fact, a disentangled or independence constraint. Non-linear ICA (Karhunen, 2001) posits that the observed signals are obtained through a nonlinear transformation of the source signals.

The variational autoencoder (VAE) (Kingma & Welling, 2022) is a modification of the autoencoder that incorporates the concept of variational inference, which can realize dimension-wise disentanglement. In VAEs, there is a term used to minimize the KL divergence between the variational posterior and the prior distribution. The chosen prior distribution is typically selected to satisfy certain inde-

**Algorithm 2** PyTorch-style Code for GMIM.

```
# f: shared neural encoder
# d: representation dimension
# adj: original graph topology
# feat: original node features
# eta: scaling factor
# epochs: total training epochs
# N: number of nodes

Id, I2d = torch.eye(d), torch.eye(2 * d)
for _ in range(epochs):
    # generate two randomly augmented views of original graph
    adj_A, feat_A = augment(adj, feat)
    adj_B, feat_B = augment(adj, feat)

    # get output representations of encoder
    H_tilde_A = f(adj_A, feat_A)
    H_tilde_B = f(adj_B, feat_B)

    # normalize representations along sample direction
    H_A = (H_tilde_A - H_tilde_A.mean(0)) / H_tilde_A.std(0)
    H_B = (H_tilde_B - H_tilde_B.mean(0)) / H_tilde_B.std(0)

    # compute covarance matrix
    Cov_A = torch.mm(H_A.T, H_A) / N
    Cov_B = torch.mm(H_B.T, H_B) / N
    CrossCov = torch.mm(H_A.T, H_B) / N
    JointCov = torch.cat([torch.cat([Cov_A, CrossCov], dim=1),
                          torch.cat([CrossCov.T, Cov_B], dim=1)], dim=0)

    # calculate loss function
    loss = torch.log(torch.det(I2d + eta * JointCov) / torch.det(Id + eta * Cov_A)
              / torch.det(Id + eta * Cov_B))

    # update parameters
    loss.backward()
    optimizer.step()
```

pendent properties, such as an isotropic Gaussian distribution. As a result, the KL divergence term potentially imposes a independent constraint on the latent variables. The $\beta$-VAE (Higgins et al., 2017) multiplies the KL divergence term by a penalty factor $\beta$ to enhance the disentangling effect on the latent variables. The KL divergence term shares a similar underlying principle with the entropy maximization term in our paper. Chen et al. (2018) demonstrate that the penalty term in $\beta$-VAE tends to enhance the dimension-wise independence of the latent variable, but it also diminishes the capacity of the latent variables to preserve information from the input. Similar to decorrelation-based self-supervised learning methods such as CCA-SSG, DIP-VAE (Kumar et al., 2018) directly regularizes the elements in the covariance matrix of the posterior distribution, making it approach the identity matrix. FactorVAE (Kim & Mnih, 2018) introduces a term known as Total Correlation to quantify the level of dimension-wise independence.

## G    ALGORITHM

The overall algorithm flows for GMIM and GMIM-IC in the form of PyTorch-style pseudocode are placed in Algorithm 2 and 3, respectively.

## H    MORE EXPERIMENTS AND STATISTICS OF DATASETS

### H.1    EFFICIENCY COMPARISON

**Efficiency Comparison.**    To illustrate the simplicity and efficiency of our model, we compare our method with other graph contrastive methods based on mutual information estimators in terms of numbers of model parameters, time consumption of training stage, and memory costs. Table 2 summarizes all indicators of various methods. Overall, compared to other methods, our method has fewer model parameters, shorter training time, and smaller memory costs in most cases. This is because our method doesn't rely on additional projection heads, parameterized mutual information estimator, and negative samples, which add extra calculation, additional parameters, and storage burden. Besides, the short training time potentially indicates the fast convergence of our algorithm.

**Algorithm 3** PyTorch-style Code for GMIM-IC.

```python
# f: shared neural encoder
# d: representation dimension
# adj: original graph topology
# feat: original node features
# eta: scaling factor
# beta: balancing factor
# epochs: total training epochs
# N: number of nodes

Id = torch.eye(d)
for _ in range(epochs):
    # generate two randomly augmented views of original graph
    adj_A, feat_A = augment(adj, feat)
    adj_B, feat_B = augment(adj, feat)

    # get output representations of encoder
    H_tilde_A = f(adj_A, feat_A)
    H_tilde_B = f(adj_B, feat_B)

    # normalize representations along sample direction
    H_A = (H_tilde_A - H_tilde_A.mean(0)) / H_tilde_A.std(0)
    H_B = (H_tilde_B - H_tilde_B.mean(0)) / H_tilde_B.std(0)

    # compute covarance matrix
    Cov_A = torch.mm(H_A.T, H_A) / N
    Cov_B = torch.mm(H_B.T, H_B) / N

    # calculate loss function
    loss_ic = (H_A - H_B).pow(2).sum() / N
    loss_em = - torch.log(torch.det(Id + eta * Cov_A) * torch.det(Id + eta * Cov_B))
    loss = loss_ic + beta * loss_em

    # update parameters
    loss.backward()
    optimizer.step()
```

Table 2: Comparison of numbers of model parameters, training time, and memory costs between various graph contrastive methods.

| Algorithm | Cora | | | Citeseer | | | Pubmed | | | Computers | | |
|---|---|---|---|---|---|---|---|---|---|---|---|---|
| | Paras | Time | Memory | Paras | Time | Memory | Paras | Time | Memory | Paras | Time | Memory |
| DGI | 996K | 6.8s | 3.8GB | 2158K | 9.4s | 7.8GB | 194K | 44.9s | 11.2GB | 1,808K | 71.2s | 11.3GB |
| GRACE | 433K | 5.1s | 1.2GB | 2,159K | 7.4s | 1.5GB | 519K | 1,169s | 12.2GB | 263K | 362.8s | 7.4GB |
| MVGRL | 1,731K | 23.7s | 3.8GB | 4,055K | 48.4s | 7.9GB | 322K | 2,010s | 9.1GB | 1,049K | 78.8s | 16.6GB |
| GMIM | 997K | 2.8s | 2.5GB | 1,896K | 2.5s | 2.6GB | 289K | 9.1s | 2.8GB | 656K | 7.5s | 3.2GB |
| GMIM-IC | 997K | 3.1s | 2.5GB | 1,896K | 2.9s | 2.6GB | 289K | 6.6s | 2.8GB | 656K | 8.7s | 3.2GB |

The simplicity of our model and the efficiency of the calculation of objective function significantly reduce the time and space complexity of our method.

## H.2 VISUALIZATION OF CORRELATION MATRIX

Figure 8 provides visualizations of correlation matrices of node representations under various settings on Cora and Pubmed. Specifically, for a representation matrix $\mathbf{H} \in \mathbb{R}^{N \times d}$ which has been normalized to 0-mean and 1-standard deviation, the correlation matrix is $\frac{1}{N}\mathbf{H}^\top \mathbf{H}$. In other words, each element of correlation matrix denotes the Pearson correlation coefficient of two variables (*i.e.*, two channels). As shown in Figure 8(a,d), the off-diagonal elements of correlation matrices are large without considering entropy maximization term in GMIM-IC, indicating that various channels of representation matrix coupled together. That is to say, the issue of dimensional collapse has occurred. Moreover, the two proposed variants, GMIM and GMIM-IC, can effectively decorrelate various representation channels and mitigate the dimensional collapse issue.

## H.3 SYNERGISTIC EVOLUTION BETWEEN GMI AND THAT WITH SHIFTED AND SCALED EIGENVALUES

In the main context, considering that directly designing the objective function based on Gaussian mutual information $I_G(X;Y) = \frac{1}{2}\ln \frac{\det(\mathbf{\Sigma}_X)\cdot\det(\mathbf{\Sigma}_Y)}{\det(\mathbf{\Sigma}_{X,Y})}$ will lead numerical instability, we proposed a feasible alternative by shifting and scaling the eigenvalues of the covariance matrix, denoted as $I'_G(X;Y) = \ln \frac{\det(\mathbf{I}+\eta\cdot\mathbf{\Sigma}_X)\cdot\det(\mathbf{I}+\eta\cdot\mathbf{\Sigma}_Y)}{\det(\mathbf{I}+\eta\cdot\mathbf{\Sigma}_{X,Y})}$.

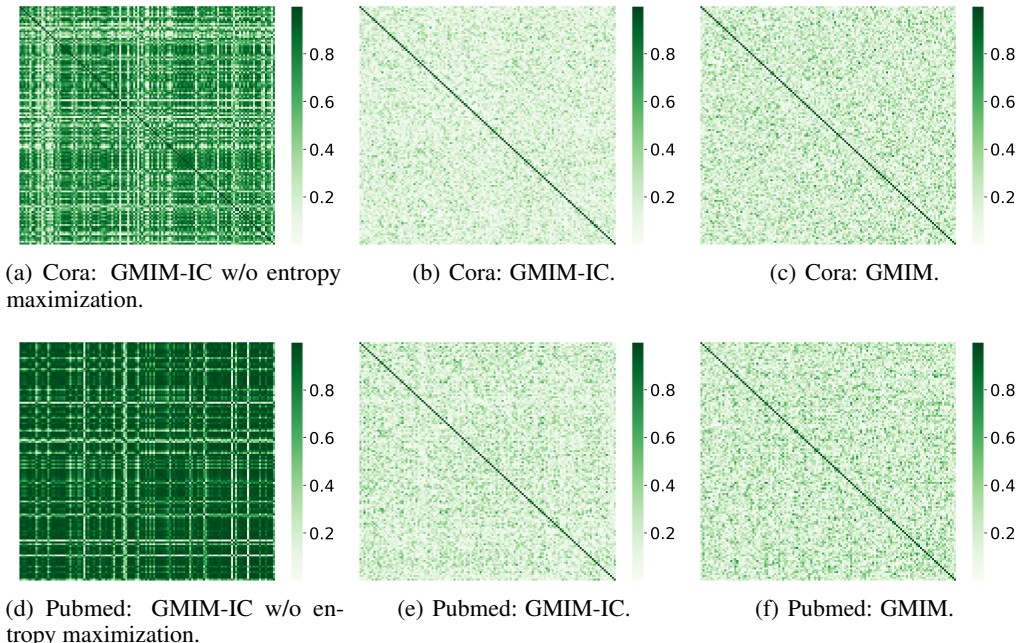

(a) Cora: GMIM-IC w/o entropy maximization.

(b) Cora: GMIM-IC.

(c) Cora: GMIM.

(d) Pubmed: GMIM-IC w/o entropy maximization.

(e) Pubmed: GMIM-IC.

(f) Pubmed: GMIM.

Figure 8: Visualizations of the correlation matrices (absolute value) of representations under various settings on Cora and Pubmed.

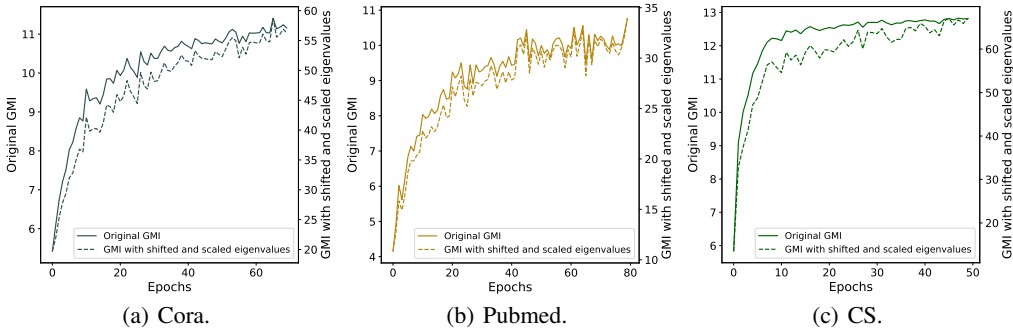

(a) Cora.

(b) Pubmed.

(c) CS.

Figure 9: Synergistic evolution between GMI and that with shifted and scaled eigenvalues. Specifically, the original GMI (solid line) is adopted as the optimization objective while the corresponding values (dashed line) of GMI with shifted and scaled eigenvalues are recorded.

We visualize the synergistic evolution between $I_G$ and $I'_G$ during the training phase in Figure 9. It is worth mentioning that all experiments are conducted under low-dimensional settings, where the original GMI $I_G$ remains normal and valid values. It can be observed that the two lines exhibit consistent patterns of variation in each subfigure. This phenomenon provides evidence supporting the rationality of obtaining a viable objective function through shifting and scaling the eigenvalues.

## H.4 EFFECT OF AUGMENTATION INTENSITY

We conduct a sensitivity analysis on the augmentation intensity by examining the effects of various combinations of the edge removal ratio $p_e$ and the feature masking ratio $p_f$. The results, presented in Figure 10, indicate that our method is more sensitive to augmentation in features ($p_f$) compared to that in graph structure ($p_e$). Overall, within an appropriate range of $p_e$ and $p_f$, our approach consistently achieves competitive results. Even when subjected to strong augmentation (*e.g.*, $p_e = 0.6$ and $p_f = 0.6$), our method still maintains a satisfactory performance level.

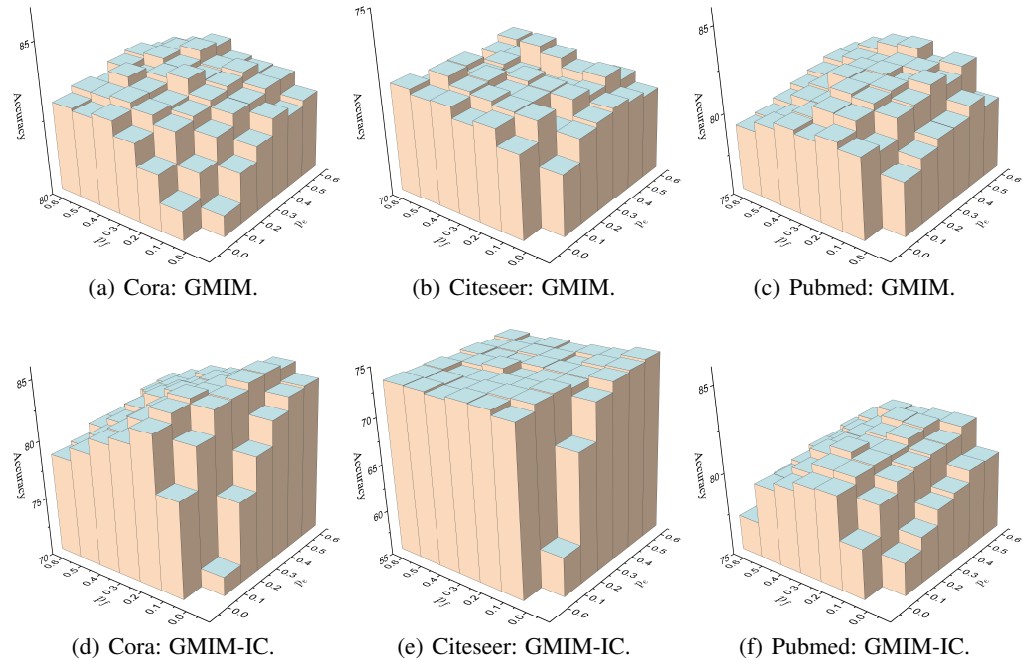

(a) Cora: GMIM.  (b) Citeseer: GMIM.  (c) Pubmed: GMIM.

(d) Cora: GMIM-IC.  (e) Citeseer: GMIM-IC.  (f) Pubmed: GMIM-IC.

Figure 10: The classification accuracy under various combinations of feature masking ratio $p_f$ and edge removal ratio $p_e$.

## H.5 T-SNE VISUALIZATIONS.

To achieve a more profound understanding of our approach and conduct a comprehensive comparison with different methods, a series of t-SNE plots (Van der Maaten & Hinton, 2008) is employed to visualize the raw features and learned representations under various methods and distinct configurations in Figure 11. In Figure 11(a), the 2-dimensional t-SNE embeddings of the raw features present a chaotic distribution and can not show discriminative clusters. The visualization in Figure 11(b), characterized by a complex elliptical shape, highlights that the method lacking the identity constraint term can not capture semantically meaningful information. in Figure 11(c), the two dimensions of the t-SNE embeddings show a certain correlation, potentially indicating dimensional collapse issue in high-dimensional representation space. This phenomenon illustrates the effect of the entropy maximization term in learning diverse representations and avoiding dimensional collapse. In Figure 11(d), the t-SNE results form discernible and interpretable clusters based on their true categories, indicating that our method can learn meaningful and diverse representations. The second row in Figure 11 shows t-SNE embeddings of the other four methods, and the visual results of various methods do not exhibit significantly distinct appearance. However, upon closer inspection, our method demonstrates better inter-class discriminability, especially concerning the clusters in purple, green, and blue.

## H.6 EXPERIMENTS ON OGBN-ARXIV

To further evaluate the effectiveness and efficiency of our method, we conduct experiments on a large-scale graph Ogbn-Arxiv (Hu et al., 2020). Table 3 reports the validation and test accuracy of various graph self-supervised methods, where our method obtains good performance. It is worth mentioning that GRACE and GCA do not operate on a full graph manner but a subset of nodes are sampled as negative samples to avoid memory issues. Moreover, Figure 12 simultaneously presents the test accuracy and training time, indicating that our method can effectively balance performance and efficiency.

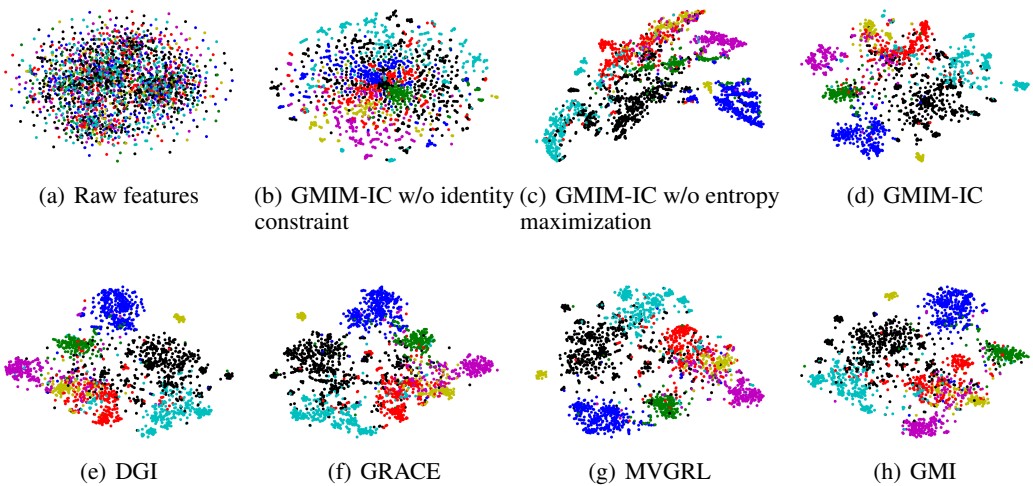

(a) Raw features    (b) GMIM-IC w/o identity constraint    (c) GMIM-IC w/o entropy maximization    (d) GMIM-IC

(e) DGI    (f) GRACE    (g) MVGRL    (h) GMI

Figure 11: t-SNE visualizations of the raw features and learned representations of various methods on Cora. "w/o" stands for "without". Best viewed in colors.

Table 3: Validation and test accuracy on Ogbn-Arxiv. "OOM" indicates out-of-memory on a GPU with 24GB memory.

|  | Validation | Test |
|---|---|---|
| DGI | $71.19 \pm 0.24$ | $70.28 \pm 0.23$ |
| MVGRL | OOM | OOM |
| GMI | OOM | OOM |
| CCA-SSG | $72.35 \pm 0.17$ | $71.33 \pm 0.21$ |
| BGRL | $72.58 \pm 0.14$ | $71.52 \pm 0.14$ |
| GRACE | $71.82 \pm 0.18$ | $70.91 \pm 0.21$ |
| GCA | $71.63 \pm 0.20$ | $70.77 \pm 0.22$ |
| GMIM (ours) | $72.26 \pm 0.16$ | $71.27 \pm 0.21$ |
| GMIM-IC (ours) | $72.48 \pm 0.18$ | $71.42 \pm 0.19$ |

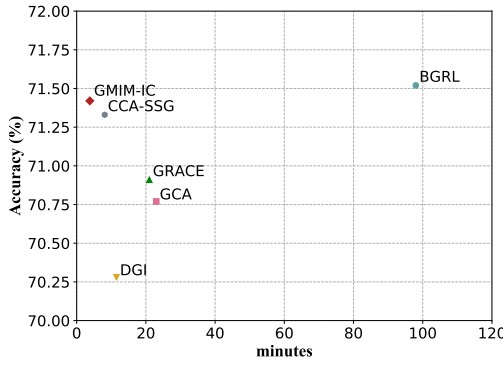

Figure 12: Classification accuracy in test set and training time on Ogbn-Arxiv.

## H.7 DETAILS OF HYPERPARAMETER CONFIGURATION

The details of hyperparameter configuration for GMIM and GMIM-IC are placed in Table 4 and 5, respectively.

## H.8 DETAILS OF THE EXPERIMENTAL DATASETS

The statistics of the experimental datasets are summarized in Table 6. The details of the datasets are as follows:

- **Cora, Citeseer,** and **Pubmed** are citation networks where nodes represent documents and edges denote citation relationships. Each document is assigned a class label that indicates its subject category, and it is characterized by a bag-of-words feature vector.

- **Amazon-Computers** and **Amazon-Photo** are two graphs derived from the Amazon dataset, capturing co-purchase relationships. The nodes in these graphs represent products, and an edge exists between two nodes if they are frequently purchased together. Each node is associated with a sparse bag-of-words feature vector based on product reviews. The category of each node is indicated by its label.

Table 4: Hyperparameter configuration of the experiments for GMIM. "lr" indicates learning rate while "wd" denotes weight decay.

| Dataset | GMIM | | | | | |
| | Layers | Representation dim | lr | wd | $p_f$ | $p_e$ |
|---|---|---|---|---|---|---|
| Cora | 2 | 512 | 1e-3 | 0 | 0.4 | 0.5 |
| Citeseer | 1 | 512 | 1e-3 | 0 | 0.4 | 0.5 |
| Pubmed | 2 | 64 | 1e-3 | 0 | 0.2 | 0.6 |
| Computers | 2 | 512 | 1e-3 | 0 | 0.1 | 0.3 |
| Photo | 2 | 512 | 1e-3 | 0 | 0.2 | 0.3 |
| Coauthor-CS | 2 | 512 | 1e-3 | 0 | 0.2 | 1.0 |

Table 5: Hyperparameter configuration of the experiments for GMIM-IC.

| Dataset | GMIM-IC | | | | | | |
| | Layers | Representation dim | $\beta$ | lr | wd | $p_f$ | $p_e$ |
|---|---|---|---|---|---|---|---|
| Cora | 2 | 512 | 1.0 | 1e-3 | 0 | 0.1 | 0.5 |
| Citeseer | 1 | 512 | 0.5 | 1e-3 | 0 | 0.0 | 0.6 |
| Pubmed | 2 | 64 | 3.0 | 1e-3 | 0 | 0.3 | 0.5 |
| Computers | 2 | 512 | 4.0 | 1e-3 | 0 | 0.1 | 0.3 |
| Photo | 2 | 512 | 7.0 | 1e-3 | 0 | 0.2 | 0.3 |
| Coauthor-CS | 2 | 512 | 1.0 | 1e-3 | 0 | 0.2 | 1.0 |

- **Coauthor-CS** is an academic network in the field of computer science, where nodes represent authors and edges indicate co-authorship relationships. Two authors are connected by an edge if they have collaborated on a research paper.

- **Ogbn-Arxiv** is a directed citation network among some computer science arXiv papers. Each node on the graph corresponds to an arXiv paper, while directed edges indicate the citing relationships between papers. Each paper is associated with a 128-dimensional feature vector.

Table 6: Statistics of the experimental datasets.

| Dataset | Nodes | Edges | Features | Classes |
|---|---|---|---|---|
| Cora | 2,708 | 5,429 | 1,433 | 7 |
| Citeseer | 3,327 | 4,732 | 3,703 | 6 |
| Pubmed | 19,717 | 44,338 | 500 | 3 |
| Amazon-Computers | 13,752 | 245,861 | 767 | 10 |
| Amazon-Photo | 7,650 | 119,081 | 745 | 8 |
| Coauthor-CS | 18,333 | 81,894 | 6,805 | 15 |
| Ogbn-Arxiv | 169,343 | 2,332,386 | 128 | 40 |

