# OpenReview forum: "Gaussian Mutual Information Maximization for Graph Self-supervised Learning: Bridging Contrastive-based to Decorrelation-based"
_ICLR.cc/2024/Conference — Submitted to ICLR 2024_

### Official Review · Reviewer_jkcF · 2023-10-25

**Soundness:** 2 fair
**Presentation:** 3 good
**Contribution:** 2 fair
**Rating:** 3
**Confidence:** 4

**Summary:**

The authors introduce a new model for self-supervised learning of contrastive representations on graphs. The latter focuses on instance-level discrimination as a recent model based on Canonical Correlation Analysis (CCA-SSG), which has been shown to avoid several problems of previous contrastive learning methods, such as sample augmentation efficiency and dimension collapse. Instead, the authors base their analysis on mutual information (MI) under Gaussian assumptions that benefit from closed-form solutions that avoid complex estimators of MI as is often the case in this literature. Thus, they introduce a first contrastive learning approach for Gaussian MI and two other variants to overcome some computational limitations. They then explicit certain theoretical links with the CCA-SSG model. Finally, they conclude with a benchmark of SSL methods for node classification tasks and several ablation studies that show the competitiveness of their methods.

**Strengths:**

-	Overall the paper is well-written, the methods and results are clearly presented.
-	Propose several new models with different contrastive loss functions: i) a Gaussian Mutual Information (IG) loss that depends on determinant of covariance matrices over augmented views – ii) A transformed IG loss (GMIM) that adds an offset and scaling to these covariance matrices to overcome stability issues of IG as determinants converge to zero when seeking for decorrelated dimensions – iii) GMIM-IC which results from a reformulation of GMIM via conditional entropies further assumed to reduce to the identity.
-	Authors evaluate their approach by learning node embeddings using GMIM in an unsupervised setting. Then evaluate their embeddings in a supervised way on node-classification tasks using logistic regressions. In this framework, GMIM and GMIM-IC performances compete or outperform over SSL approaches.
-	Conduct several additional analysis of interest : (main paper) effect of representation dimension, balancing coefficients for GMIM-IC, cumulated losses synergy and (supplementary) feature and edge removal probabilities.

**Weaknesses:**

**Overall appreciation**:  The empirical and theoretical analysis is clearly a continuation of previous work. The main contributions seem to be ways of stabilizing the learning of these models by introducing certain biases. This is clearly necessary for more advanced reflections in the future. However, as it stands, these considerations seem to me to be under-exploited and would greatly benefit from the introduction of several current reflections on the GNN literature.

1.	**Redundant theoretical analysis**: The theoretical contributions seem rather straight-forward as most can be found or deduced from other papers without these being clearly mentioned.
-    a)	Mutual Information between multivariate Gaussians is already well-studied and can be found in several classical books of Information Theory & Statistics. As it clearly connects with closed-form solutions for the KL divergence e.g [A], [B] and references therein.
-    b)	The relations and differences to the CorInfoMax paper should be clearly stated in terms of (theoretical) design to provide a more comprehensive overview of MI variants. As indeed CorInfoMax relies on log-determinant MI (LDMI) which directly connects with the entropy of gaussian distributions. Differences with GMIM mostly lie in the way to handle instability with different offset and scaling techniques. Note that some theoretical analysis of LDMI-based papers also overlap with some results in the paper.
-    c)	Analysis of IG-based solutions clearly connect with the information theory point of view of CCA-SSG already developed in this paper.

2.	**Missing points in the experiments**:
   - a)	There are no clear descriptions of the architectures, validated hyperparameters / best configurations reported in Table 1. This prevents us from accessing the fairness of these benchmarks. Please could you provide those ?
   - b)	Gains in terms of performances seem rather marginal CCA-SSG. As briefly mentioned in the conclusion, it would have been relevant to explore different prior distributions.
   - c)	lack of clarity or hindsight w.r.t the evaluation : No clear justifications for the choice of supervised evaluation. No experiments in semi-supervised settings. No fully unsupervised evaluations, e.g using KL-based clustering methods such as KL-quantization methods.
   - d)	No sensitivity analysis w.r.t the encoder, I guess a GNN backbone. Nor a clear comparison between performances of this GNN backbone in a fully supervised setting vs the 2-step strategy used by authors to evaluate GMIM embeddings. Such analysis could relate to the common concerns in the GNN literature e.g i) expressivity simply considering e.g several GNN layers using Jumping Knowledge based backbones; ii) homophily vs heterophily via e.g [C] whose supervised models exhibit much higher classification performances than those reported in Table 1.
   - e)	Incomplete or arguable analysis of the results.

[A] Bouhlel, N., & Dziri, A. (2019). Kullback–Leibler divergence between multivariate generalized gaussian distributions. IEEE Signal Processing Letters, 26(7), 1021-1025.

[B] Zhang, Y., Liu, W., Chen, Z., Wang, J., & Li, K. (2021). On the Properties of Kullback-Leibler Divergence Between Multivariate Gaussian Distributions. arXiv.

[C] Luan, Sitao, et al. "Revisiting heterophily for graph neural networks." Advances in neural information processing systems35 (2022): 1362-1375.

**Questions:**

I invite the authors to discuss the weaknesses I have mentioned above and to provide additional results/analyses for refutation, so that I can eventually increase my score. Follows some questions to clarify some points.

**Q1.** In my opinion the analysis of the dynamics w.r.t the embedding dimension is really biased by the choice of linear classifier that is used in the paper (and other papers). From my understanding, the “macro”-dynamics are the following: It is expected that in small dimensions, the embeddings have a highly non-linear geometry. Whereas the latter tends to be more linear as the dimension grows. So the linear classifier would better suit these settings but it does not mean that embeddings are more informative. Moreover, every statistical tools involved e.g empirical covariance matrices would be poorer estimates of true covariance matrices (cf classical concentration inequalities) given a fixed number of nodes while increasing the dimension. So could you

-	i) further justify this choice of linear classifier
-	Ii) provide evaluations with a non-linear classifier e.g 2-MLP with ReLU activation ?
-	iii) complete figure 2 with the other benchmarked datasets and check if there are correlations with their statistics provided in Table 4 ?

**Q2.** I am not convinced by conclusions and analysis regarding the non-gaussian behaviour of components in the embedding, and believe that it is mostly a bias that comes from the GNN backbone.

-    i ) Could you detail experiments that correspond to Figure 6 ?
-    ii) Could you also check via hypothesis testing the evolution of the gaussian behaviour w.r.t the embedding dimensions ?

**Q3.** Isotropic gaussian distributions are probably the best ones to promote uniformity in the embedding space so it does not seem so obvious that other prior distributions would perform better. Could you further discuss this point ?

---

> ### Author Response · Authors · 2023-11-19
> **Response to Reviewer jkcF (Part 1)**
>
> We sincerely appreciate your detailed, professional, and valuable reviews, which greatly contribute to the improvement of our work! Next, we will try our best to address your questions and further enhance our work.
>
> **Question 1:** The theoretical contributions seem rather straight-forward as most can be found or deduced from other papers.
>
> **Answer:** Here, we will respond point by point to the questions raised by the reviewer in the "**Redundant theoretical analysis**" part.
>
> (1) Mutual Information between multivariate Gaussians is already well-studied and can be found in several classical books of Information Theory and Statistics.
>
> *Answer:* We are grateful to the reviewer for providing the references. Indeed, as the reviewer pointed out, there have been some researches on Gaussian Mutual Information (GMI) in the fields of statistics and information theory. In our revision, we have included citations involving GMI. Simultaneously, we will preserve the derivation process of Gaussian mutual information in Appendix A to ensure the self-completeness of our paper. This allows interested readers to directly browse the derivation of GMI without searching for related literature. To be honest, we have not found any existing literature that provides a complete derivation process for Gaussian mutual information.
>
> (2) The relations and differences between our paper and CorInfoMax should be clearly stated in terms of design. The distinctions between them mainly lie in how to handle instability.
>
> *Answer:* In the design of objective function, the ways in addressing the numerical instability issue during the optimization process are a significant distinction between the two works. Besides, compared with the ways in addressing the instability, more important difference we think is that our approach does not employ additional projection heads. Our objective function directly operates on the output representations of the encoder, while CorInfoMax first employs a projector of 3-layer MLP to map the outputs of the encoder to a new embedding space and then calculate loss function in the new space. This distinction is not only the discrepancy in network architecture but, more importantly, it actually indicates the difference in motivations between the two works. The purpose of our research is to directly calculate mutual information under the Gaussian assumption without reliance on neural estimators or additional designs. *If our method could only work with the blessing of the additional projection heads, we would not complete this paper, which is contrary to our original motivation.*  In fact, we found the paper CorInfoMax after we had almost finished our work. There are indeed some similarities between the two works. Hence, we cited this paper and spent considerable space on relations and differences between the two works in Appendix F. We have done our best to make these statements comprehensive and objective.
>
> (3) The analysis of $I_G$-based solution clearly connects with the information theory point of view of CCA-SSG already developed in this paper.
>
> *Answer:* The $I_G$-based variant is obtained from the original formula of Gaussian mutual information. In our paper, the analysis about it mainly involves two aspects. Firstly, we point out that directly optimizing $I_G$-based objective will result in numerical instability and thus introduce a strategy of offsetting and scaling eigenvalues. Secondly, we decompose Gaussian mutual information maximization into two components (conditional entropy minimization and information entropy maximization). Further, we introduce identity constraint to realizing conditional entropy minimization, which aligns with the design of our shared encoder across two views. Besides, the discussion focuses on the relationship between the encoded representations from two views *A* and *B*, that is, the two variables in the formula are respectively linked to the representations from two views.
>
> Different from ours, CCA-SSG interprets its objective from the view of information theory, which focuses on explaining the roles of the two items in the objective function. The relationship discussed in CCA-SSG involves the encoded representations and the inputs of the encoder, that is, the two variables in the formula are respectively related to the inputs and outputs of the neural networks.
>
> In summary, both works have analyzed their methods from the perspective of information theory, but they differ in two aspects including purposes and objects (variables).

---

> > ### Comment · Reviewer_jkcF · 2023-11-21
> > **Answer to authors 1**
> >
> > Thank you for these detailed answers. Follows some remarks:
> > **Q1** : Indeed it seems to be the good way to go. I indeed could not find the exact GMI derivation, but it is very similar to the well-known derivations of KL divergence between gaussians and so on, so the references I previously shared are indeed relevant to position the work (as you updated in paper).
> >
> > **Q2** : Looking further into details, the references used by CorInfoMax for the Log det can be clearly discussed in your paper -> In [D], authors position well these log-det losses, discuss gaussian cases and adopt an analog offset+scaling strategy than yours. [E] and references therein also discuss relations with MI and entropy losses which can help understand relations to GMI as an upper bound to many objective functions. On the whole, it would be of good theoretical interest for the contrastive learning literature to develop these existing but scattered relationships in your paper or in the supplementary material.
> > Then I agree on your argument for the projector heads, it's not clear for me why it is needed in the framework. We also have to acknowledge that CorInfomax originally operates on images while leveraging pre-trained models.
> >
> > **Q3**: Yes I understood these differences. The gaussian assumption for CCA-SSG is done a posteriori only for their theoretical analysis, while yours is at the core of model design.
> > I am not sure about what you mean by taking into account inputs + outputs. Essentially CCA-SSG puts a prior on the conditional distribution of contrastive embeddings for a given input. If we want to be really rigorous mathematically, this dependency should be always emphasised. It is just that it would matter (e.g if you want to theoretically analyse the sensitivity to the perturbation model) or not (e.g your case) depending on what you want to prove.
> >
> > [D] Zhouyin, Zhanghao, and Ding Liu. "Understanding neural networks with logarithm determinant entropy estimator." arXiv preprint arXiv:2105.03705 (2021).
> > [E] Erdogan, Alper T. "An information maximization based blind source separation approach for dependent and independent sources." ICASSP 2022-2022 IEEE International Conference on Acoustics, Speech and Signal Processing (ICASSP). IEEE, 2022.

---

> ### Author Response · Authors · 2023-11-19
> **Response to Reviewer jkcF (Part 2)**
>
> **Question 2:** Some experimental settings and details are missing.
>
> **Answer:** Below is our response to questions regarding experimental details and configuration.
>
> (1) There are no clear descriptions of the architectures and hyperparameters for the results reported in Table 1.
>
> *Answer:* The network architectures are implemented through Graph Convolutional Networks (GCNs). The hyperparameters for GMIM are as follows, where "lr" denotes learning rate and "wd" indicates weight decay:
>
> |   Dataset   | Layers | Representation dim |  lr  |  wd  | $p_f$ | $p_e$ |
> | :---------: | :----: | :----------------: | :--: | :--: | :---: | :---: |
> |    Cora     |   2    |        512         | 1e-3 |  0   |  0.4  |  0.5  |
> |  Citeseer   |   1    |         64         | 1e-3 |  0   |  0.4  |  0.5  |
> |   Pubmed    |   2    |        512         | 1e-3 |  0   |  0.2  |  0.6  |
> |  Computers  |   2    |        512         | 1e-3 |  0   |  0.1  |  0.3  |
> |    Photo    |   2    |        512         | 1e-3 |  0   |  0.2  |  0.3  |
> | Coauthor-CS |   2    |        512         | 1e-3 |  0   |  0.2  |  1.0  |
>
> The hyperparameters for GMIM-IC are as follows:
>
> |   Dataset   | Layers | Representation dim | $\beta$ |  lr  |  wd  | $p_f$ | $p_e$ |
> | :---------: | :----: | :----------------: | :-----: | :--: | :--: | :---: | :---: |
> |    Cora     |   2    |        512         |   1.0   | 1e-3 |  0   |  0.1  |  0.5  |
> |  Citeseer   |   1    |         64         |   0.5   | 1e-3 |  0   |  0.0  |  0.6  |
> |   Pubmed    |   2    |        512         |    3    | 1e-3 |  0   |  0.3  |  0.5  |
> |  Computers  |   2    |        512         |    4    | 1e-3 |  0   |  0.1  |  0.3  |
> |    Photo    |   2    |        512         |    7    | 1e-3 |  0   |  0.2  |  0.3  |
> | Coauthor-CS |   2    |        512         |    1    | 1e-3 |  0   |  0.2  |  1.0  |
>
> (2) The gains in terms of performance seem rather marginal compared with CCA-SSG in Table 1.
>
> *Answer:* As pointed out by the reviewer, our method shows only marginal performance improvement compared to CCA-SSG. We discuss the theoretical relationship between the two methods in our paper, which implies that they may have similar experimental results. From this perspective, the experimental results presented in Table 1 can serve as an empirical support for the theoretical analysis, which we will state on in the experimental analysis of our revision.

---

> ### Author Response · Authors · 2023-11-19
> **Response to Reviewer jkcF (Part 3)**
>
> (3) The paper lacks justifications for the choice of supervised evaluation. Besides, there are no experiments in semi-supervised settings or fully unsupervised evaluations.
>
> *Answer:* After self-supervised pre-training for graph $G(\mathbf{A}, \mathbf{X})$, we obtain node representations $\mathbf{H} = [\mathbf{h}_1,...,\mathbf{h}_N]^{\top}$ and evaluate their qualities with simple linear classifier $y = \mathbf{W} x + \mathbf{b}$ under supervised settings. Good node representations should exhibit discriminability in space according to their true categories. The linear classifier under the supervised manner maps node representations to label space, and the linear classification results can well reflect the linear separability and quality of node representations. Besides, the supervised linear evaluation is a widely adopted practice in the field of self-supervised learning, including graph and computer vision. The self-supervised baselines in their papers almost universally employ linear evaluation for comparison. Thus, we also follow this practice.
>
> Here, we evaluate the learned representations on node clustering task in an unsupervised manner, where the number of categories serves as a known parameter. The clustering results are evaluated through two metrics: Normalized Mutual Information (NMI) and Adjusted Rand Index (ARI). The compared baselines include k-means, spectral clustering, DeepWalk [1], DNGR [2], MGAE [3], DAEGC [4], DGI [5], and GRACE [6]. The experimental results are as follows:
>
> |       Dataset       | Cora  | Cora  | Citeseer | Citeseer | Pubmed | Pubmed |
> | :-----------------: | :---: | :---: | :------: | :------: | :----: | :----: |
> |     **Metrics**     |  NMI  |  ARI  |   NMI    |   ARI    |  NMI   |  ARI   |
> |       k-means       | 0.377 | 0.149 |  0.241   |  0.154   | 0.263  | 0.247  |
> | spectral clustering | 0.265 | 0.158 |  0.082   |  0.075   | 0.136  | 0.091  |
> |      DeepWalk       | 0.401 | 0.254 |  0.238   |  0.085   | 0.251  | 0.203  |
> |        DNGR         | 0.318 | 0.142 |  0.180   |  0.043   | 0.153  | 0.059  |
> |        MGAE         | 0.489 | 0.436 |  0.416   |  0.425   | 0.271  | 0.224  |
> |        DAEGC        | 0.528 | 0.496 |  0.397   |  0.410   | 0.266  | 0.278  |
> |         DGI         | 0.565 | 0.529 |  0.431   |  0.434   | 0.281  | 0.263  |
> |        GRACE        | 0.556 | 0.518 |  0.377   |  0.369   | 0.251  | 0.230  |
> |       GMIM-IC       | 0.578 | 0.533 |  0.441   |  0.452   | 0.303  | 0.281  |
>
> For DeepWalk, the number of random walks is 10, the path length of each walk is 20, the context size is 10, and the number of embedding dimension is 256. For DNGR, the encoder is built with two 512-dim hidden layers and a 256-dim embedding layer. For RMSC, the trade-off parameter is set to 0.005. For MGAE, we set the number of layers, corruption level $p$, and regularization coefficient $\lambda$ to 3, 0.4, and $10^{-5}$ respectively. For DAEGC, the encoder is constructed with a 256-dim hidden layer and a 16-dim embedding layer, and the clustering coefficient is set to 10. Due to time limitation, we currently only present the results in the table. In the future, we will conduct more clustering-related experiments.
>
> (4) There is no sensitivity analysis w.r.t the encoder.
>
> *Answer:* We adopt simple Graph Convolutional Networks (GCNs) [7] as the encoder, which is stated in Section 6.1. The relevant supervised counterpart is listed in Table 1. The paper [8] adopts different splits of the datasets from ours, so we kindly argue that directly comparing the two works is somewhat unreasonable. The work [8] is a great job with good performance. However, in our study, we primarily focus on the performance improvement brought by self-supervised learning strategies rather than the backbone network itself.
>
> (5) The analysis of the results is incomplete or arguable.
>
> *Answer:* Thank you for your guidance.  In our revision, we have conducted a more detailed analysis and discussion of the experimental results.

---

> ### Author Response · Authors · 2023-11-19
> **Response to Reviewer jkcF (Part 4)**
>
> **Question 3:** The reviewer thinks that the dynamics with regard to representation dimension is biased by the choice of linear classifier. The representations tend to be more linear as the dimension grows. So the linear classifier would better suit these settings but it does not mean that embeddings are more informative. Besides, empirical covariance matrices would be poorer estimates of true covariance matrices given a fixed number of nodes while increasing the dimension.
>
> **Answer:** Many thanks to the reviewer's thoughts and advice. We fully admire your perspectives. As the dimension increases, the representations become more linearly separable in high-dimensional space, naturally contributing to improved classification results. The number of nodes (i.e., empirical observations) is fixed, and as the dimension increases, it becomes more challenging to obtain an accurate empirical estimation of the true covariance matrix, which potentially hinders the effectiveness of self-supervised training in high-dimensional settings. We sincerely appreciate the reviewer's reminder, as we did not initially consider the perspective of a poorer estimation of covariance matrix. The related analysis has been added to the experimental part w.r.t representation dimensions in our revision.
>
> (1) Can the authors further justify the choice of linear classifier?
>
> *Answer:* In general, we expect representations to cluster in space based on their true categories. The node classification task is employed to evaluate the linear separability and discriminative characteristics of representations. The linear classifier has limited expressive power, merely performing straightforward linear mapping. In this scenario, the classification performance is predominantly determined by the quality of node representations themselves. In other words, the classification results are a combined outcome of the representations themselves and the classifier, and thus the results under a simple classifier are able to effectively reflect the quality of the representations. Besides, the training set for downstream evaluation is usually small. Thus, a more complex network may lead to overfitting issues, thereby impacting the analysis and discussion of the quality of node representations themselves. Moreover, the linear evaluation is widely adopted in the field of self-supervised learning including graph and computer vision. Thus, in our paper, we follow this common practice.
>
> (2) Can the authors provide evaluations with a non-linear classifier such as 2-layer MLP with ReLU activation.
>
> *Answer:* The related experimental results are as follows:
>
> | Dataset         | Cora | Citeseer | Pubmed | Computers | Photo | CS    |
> | --------------- | ---- | -------- | ------ | --------- | ----- | ----- |
> | GMIM-IC (2-MLP) | 84.3 | 71.2     | 80.9   | 88.76     | 92.6  | 93.39 |
>
> The performance under a 2-layer MLP is weaker than that under a linear classifier. The reason is that the node representations already possess discriminative characteristics in the space, and using a linear classifier can yield satisfactory results. Applying a more complex network may actually hinder performance due to overfitting issues.
>
> (3) The authors are encouraged to complete Figure 2 with the other benchmarked datasets and check if there are correlations with their statistics of the datasets.
>
> *Answer:* In our revision, we have completed Figure 2 with the Computers dataset. The reason for not plotting $I_G$-based variant is due to encountering numerical instability issues on the Computers dataset in all conducted dimensions. Among all the datasets, Pubmed has the smallest input feature dimension. Relatively speaking, the performance on this dataset shows the most significant decline with increasing dimensions. Overall, the method is not sensitive to dimensions, especially when the representation dimension exceeds 64.

---

> ### Author Response · Authors · 2023-11-19
> **Response to Reviewer jkcF (Part 5)**
>
> **Question 4:** Some questions about the Gaussian assumption and non-Gaussian behavior of representations.
>
> (1) Can the authors detail experiments corresponding to visualizations of histograms in Figure 6?
>
> *Answer:* For the node representation matrix $\mathbf{H} \in \mathbb{R}^{N \times d}$, where $d$ is the number of representation dimensions (channels), we randomly select several (four in Figure 6) channels for visualizations. For each channel, we visualize the histograms of the $N$ samples. Besides, we can compute mean and variance from the $N$ samples. Furthermore, we visualize a curve of a Gaussian distribution with the estimated mean and variance, which can serve as a reference. The histogram and the Gaussian curve exhibit a certain degree of alignment, leading us to initially think that node representations follow a Gaussian distribution. However, rigorous hypothesis testing refutes our assumption. In our revision, we have added histogram visualizations for three additional datasets. On the CS and Computers datasets, the alignment between the Gaussian curve and the histogram is relatively poor.
>
> (2) Could the authors also check via hypothesis testing the evolution of the Gaussian behavior w.r.t the embedding dimensions?
>
> *Answer:* For the node representation matrix $\mathbf{H} \in \mathbb{R}^{N \times d}$, we conduct hypothesis testing separately for each channel and take the average of the results from all channels as the final testing result of the entire representation matrix. The relationships between the results of hypothesis testing and representation dimensions are as follows:
>
> | dimension | 32        | 64       | 128       | 256       | 512       | 1024      |
> | --------- | --------- | -------- | --------- | --------- | --------- | --------- |
> | Cora      | 2.52e-20  | 6.51e-8  | 1.48e-22  | 1.3e-17   | 2.48e-12  | 1.0e-13   |
> | Pubmed    | 1.48e-118 | 1.48e-64 | 1.21e-84  | 1.69e-70  | 1.33e-57  | 1.07e-63  |
> | CS        | 0.0       | 0.0      | 0.0       | 0.0       | 0.0       | 0.0       |
> | Computers | 5.45e-182 | 0.0      | 1.25e-281 | 1.94e-178 | 4.39e-285 | 5.18e-181 |
>
> A larger value indicates that the representations are closer to a Gaussian distribution. The results of the hypothesis testing do not appear to have a significant relationship with representation dimensions.
>
> **Question 5:** Isotropic Gaussian distributions are probably the best ones to promote uniformity in the embedding space so it does not seem so obvious that other prior distributions would perform better. Could the authors further discuss this point?
>
> **Answer:** We highly agree with the reviewer's opinion. We also do not think that other prior distributions would perform better than the Gaussian distribution. As a basic knowledge in statistics, the Gaussian distribution is the most common distribution in life and nature. Besides, the Gaussian assumption is justifiable and extensively employed in numerous disciplines such as economics, data science, and physics. More importantly, although the hypothesis test indicates that node representations do not exhibit a purely Gaussian distribution, the visualization results suggest that they present a Gaussian appearance. This phenomenon contributes to the study under the Gaussian assumption. Considering the above factors, in terms of performance and practical value, we are not optimistic that other priors can outperform the Gaussian distribution. However, deriving a closed-form solution for mutual information under specific distributions is not an effortless task. Therefore, please forgive us for not being able to provide empirical results for other prior distributions here. These deserve further exploration in a new research. In fact, the derivation of Gaussian mutual information has been a laborious endeavor in our paper, even though it might be available in existing literature. In the *future work* section of our paper, we also demonstrate the prospect of extending the Gaussian assumption to other priors, aiming to inspire interested readers.
>
>
>
> [1] Deepwalk: Online Learning of Social Representations. SIGKDD, 2014.
>
> [2] Deep Neural Networks for Learning Graph Representations. AAAI, 2016.
>
> [3] Mgae: Marginalized Graph Autoencoder for Graph Clustering. CIKM, 2017.
>
> [4] Attributed Graph Clustering: A Deep Attentional Embedding Approach. IJCAI, 2019.
>
> [5] Deep Graph InfoMax. ICLR, 2018.
>
> [6] Deep Graph Contrastive Representation Learning.
>
> [7] Semi-supervised classification with graph convolutional networks. ICLR, 2017.
>
> [8] Revisiting heterophily for graph neural networks. NeurIPS, 2022.
>
>
>
>
>
> Thank you again for your efforts and constructive comments. The above is our response to your questions. If we have any misunderstandings regarding your questions, please kindly point them out. We are looking forward to further communication with you.

---

> ### Comment · Reviewer_jkcF · 2023-11-21
> **Answer to authors 2**
>
> - **part 2**. Thank you for these results which should be reported in the supplementary material. Looking at the experimental details of CCA-SSG, the benchmark seems indeed fair.
>
> - **part 3**. Thank you for this fully unsupervised analysis which also supports the use of your approach in this context hence can be put in the supplementary material and referred to in the main paper.
>
> - **part 4**: Thank you for your detailed answers.
>       - For the covariance matrix estimations a good survey can be found in [F] and references therein also specify concentration bounds, essentially in $O(\sqrt{d / n})$. Interestingly, CorInfoMax took these estimation matters into account by designing a recursive estimation across batches.
>       - Just to be sure, for the provided results with a 2-MLP, did you follow the same supervised validation than for the 1-MLP results reported in the paper ?
>
> [F] Ledoit, Olivier, and Michael Wolf. "The power of (non-) linear shrinking: A review and guide to covariance matrix estimation." Journal of Financial Econometrics 20.1 (2022): 187-218.
>
> **Overall appreciation** : I find authors' rebuttal compelling so I will increase my score from 3 to 5. I emphasize that I appreciate this paper main message 'fit well in rigorous but simple theoretical framework and it will work better'. However, as discussed in [D] the approach is actually more general than presented by the authors, so the paper could be made quite richer with a better literature review. Finally, the sensitivity to the backbone (not just in terms of performances but also dynamics) as I suggested in my initial review seems essential to me in order to access how relevant this main message is.

---

> > ### Author Response · Authors · 2023-11-22
> > **Response to Reviewer jkcF During Discussion (Part 2)**
> >
> > **Feedback 3:** What did the authors mean by taking into account inputs + outputs when talking about CCA-SSG?
> >
> > **Answer:** What we are discussing here is the difference in expression form and variables between CCA-SSG and ours. We utilize Gaussian mutual information $I(X;Y) = H(X) - H(X|Y)$ to design the objective function, where $X$ and $Y$ correspond to the encoded representations from two views $A$ and $B$. In other words, they are both related to the **outputs** of the encoder. In the paper of CCA-SSG, the authors utilize $H(Z_S | X)$ and $H(Z_S)$ to analyze their objective, where $X$ represents input data of the encoder and $Z_S$ is the encoded representations. In other words, $X$ and $Z_S$ are related to the **inputs and outputs** of the encoder, respectively.
> >
> > **Feedback 4:** For the provided results with a 2-MLP, did the authors follow the same supervised validation than for the 1-MLP results reported in the paper?
> >
> > **Answer:** Yes, the supervised validation is same.
> >
> >
> >
> > **Feedback 5:** The experimental settings and details should be added to the supplementary materials. The sensitivity to the backbone (not just in terms of performances but also dynamics) seems essential to the reviewer in order to access how relevant this main message is. Besides, the unsupervised evaluations can be put in the supplementary material and referred to in the main paper.
> >
> > **Answer:** Thank you for your suggestions! We have added the experimental settings to Appendix H.7 of our revision. For our GMIM-IC, we explore the sensitivity to the backbone and some experimental results are as follows:
> >
> > |  Backbone   |      Cora      |    Citeseer    |     Pubmed     |
> > | :---------: | :------------: | :------------: | :------------: |
> > | 1-layer GCN | 82.8 $\pm$ 0.5 | 73.7 $\pm$ 0.4 | 80.3 $\pm$ 0.5 |
> > | 2-layer GCN | 84.6 $\pm$ 0.5 | 72.2 $\pm$ 1.0 | 81.8 $\pm$ 0.6 |
> > | 3-layer GCN | 82.2 $\pm$ 0.7 | 71.4 $\pm$ 1.2 | 81.6 $\pm$ 0.5 |
> > |     GAT     | 84.2 $\pm$ 0.6 | 73.9 $\pm$ 0.4 | 81.0 $\pm$ 0.3 |
> > | JK-Net [4]  |      TODO      |      TODO      |      TODO      |
> > | ACM-GCN [5] |      TODO      |      TODO      |      TODO      |
> >
> > From the table above, we can observe that both an excessive and insufficient number of layers are unfavorable for learning good representations during self-supervised training process. Inadequate network depth results in limited modeling capacity, while an excessive number of graph convolutional layers could lead to the issue of over-smoothing, resulting in performance degradation. On Citeseer dataset, the performance based on GAT outperform that based on GCN. This result is reasonable as GAT has stronger expressive power. However, on the other two datasets, the performance based on GAT is relatively weaker. We tentatively attribute this result to the inadequate tuning of hyperparameters. Firstly, all hyperparameter settings are directly adopted from the experiments based on GCN. Secondly, GAT involves hyperparameters such as the number of attention heads that also require careful fine-tuning. In terms of this point, we will conduct more detailed experiments and analysis in the following days. Besides, we also plan to perform experiments with the other two methods [4, 5] as backbone. However, since the deadline of discussion is approaching, we are currently only able to present the results shown in the above table. In the following days, we will conduct comprehensive, careful, and rigorous experiments and analysis and add the complete experimental results and analysis into our revised version even if it could not be updated on the website.
> >
> > Similarly, for unsupervised evaluation, there are other methods (GCA, GraphMAE, GMI and so on) that need to be included in the baselines for a more comprehensive comparison and assessment . We will also incorporate these contents in our revised version.
> >
> > [4] Representation learning on graphs with jumping knowledge networks. ICML, 2018.
> >
> > [5] Revisiting heterophily for graph neural networks. NeurIPS, 2022.
> >
> >
> >
> > **Feedback 6:** For the covariance matrix estimation, a good survey can be found in [6].
> >
> > **Answer:** We sincerely appreciate the thoughtful suggestions of the reviewer, which is really heartwarming! Improving the estimation of the covariance matrix will enhance the performance of our method. We plan to investigate this aspect in our future work and expansion. We have cited [6] in the final section of our revision, hoping that it can also inspire interested readers.
> >
> > [6] The power of (non-) linear shrinking: A review and guide to covariance matrix estimation. Journal of Financial Econometrics, 2022.

---

> > ### Author Response · Authors · 2023-11-22
> > **Response to Reviewer jkcF During Discussion (Part 3)**
> >
> > The above content encompasses our response to your feedback and the tasks (including literature review, experiments on various backbones and evaluations under unsupervised settings) we will undertake in the coming days. We would like to express our gratitude once again for your efforts during the review process. After multiple rounds of discussion and communication, we believe that the quality of our paper has been improved. If you have any further suggestions to enhance our paper, please do not hesitate to tell us!

---

> ### Author Response · Authors · 2023-11-22
> **Response to Reviewer jkcF During Discussion (Part 1)**
>
> Dear Reviewer jkcF,
>
> We sincerely appreciate your active discussion and detailed feedback! Here are our responses to the key points raised in your feedback (including overall appreciation):
>
> **Feedback 1:** The references [2, 3] used by CorInfoMax [1] for the Log det can be clearly discussed in the authors' paper.
>
> **Answer:** Thank you for your suggestions! Here, we would first demonstrate the differences between our method and the mentioned approaches. Subsequently, we will outline how we will cite and discuss these methods in our revised version.
>
> The paper [2] also utilizes entropy estimation under the Gaussian distribution, referred to as the LogDet estimator in their work. The paper [2] primarily focuses on the interpretability of training and dynamic process of neural networks, which utilizes entropy or mutual information to assess the relationships between the outputs of different network layers, where the entropy estimator is used as **a metric (or indicator)** rather than an objective function. In contrast, we utilize Gaussian mutual information as an optimization objective for supervising the training of neural networks. [2] can be regarded as the application of information theory to the interpretability of deep learning, with a specific focus on investigating the information bottleneck theory in neural networks, while we explore self-supervised graph representation learning. Indeed, the formula in the paper [2] bears some resemblance to ours. However, it has different origins and parameter settings from ours. According to the second paragraph on page 3 of the paper [2], the identity matrix $I$ in Eq. (4) is considered as noise added from a standard Gaussian distribution. Besides, the expanding factor $\beta$ is used to **enlarge** the original covariance matrix and decrease the comparable weight of added noise according to the second paragraph on page 3. As described in our paper, we obtain our objective from the perspective of offsetting and scaling the eigenvalues. In the second paragraph of page 6 of the paper [2], the $\beta$ is set to $100d$ where $d$ is the dimension, which is a very large number. In our paper, the scaling factor $\eta$ is typically set to 0.1 to stabilize the training process. We observed that a large value of factor $\eta$ will lead to training instability, as it results in a significant increase of the determinant of the covariance matrix.
>
> The works [2, 3] can be viewed as the application of information theory in different fields, involving *the interpretability (information bottleneck theory) of neural networks* and *blind source separation (independent component analysis)*. From these two aspects, we will make a literature review and add it to our revised version.
>
> [1] Self-supervised learning with an information maximization criterion. NeurIPS, 2022.
>
> [2] Understanding neural networks with logarithm determinant entropy estimator. 2021.
>
> [3] An information maximization based blind source separation approach for dependent and independent sources. ICASSP, 2022.
>
> **Feedback 2:** Some viewpoints about projection heads in CorInfoMax and further statements regarding our paper.
>
> **Answer:** From a motivational perspective, CorInfoMax aims to mitigate the collapse issue by employing the objective based on LogDet of the covariance matrix. Furthermore, it directly replaces the commonly used InfoNCE loss in the embedding space (that is, the output space of the projection heads) with the new objective. The purpose of our research is to enable the direct calculation of mutual information under the Gaussian assumption without relying on estimators and extra projection heads. The projectors in CorInfoMax completely deviate from our initial motivation. Our approach reduces model complexity and enhances efficiency by directly optimizing the output space of the encoder. The advantage in efficiency is a notable strength that should not be overlooked. The distinction in projection heads is not only reflected in the variations in network architecture but, more importantly, it actually indicates the disparities in motivations and underlying concepts between the two works. In fact, if CorInfoMax did not adopt additional projection heads, we would not regard our proposed method based on Gaussian mutual information maximization as one of our contributions.
>
> Besides, our contributions go beyond only proposing a self-supervised method based on Gaussian mutual information maximization. It is an important issue to establish theoretical connections and a unified perspective for different self-supervised learning methods, which has received relatively limited research attention. Our study contributes to this aspect and provides insights into bridging various self-supervised learning methods, which takes up a considerable portion of our paper.

---

### Official Review · Reviewer_r9Qc · 2023-10-30

**Soundness:** 3 good
**Presentation:** 3 good
**Contribution:** 3 good
**Rating:** 6
**Confidence:** 5

**Summary:**

This paper presents Gaussian Mutual Information (GMI) as a computationally tractable surrogate for mutual information in graph contrastive learning. GMI eliminates the need for additional parameters and improves computational efficiency compared to conventional methods. The authors also highlight the connection between contrastive-based and decorrelation-based SSL methods, bridging the gap between the two research areas. Extensive experiments and empirical evidence support the effectiveness and efficiency of GMI.

**Strengths:**

1. The paper is well-written and easy to understand the findings.
2. The authors present complete and accurate derivation process of GMI based on the commonly used Gaussian assumption for node representation. Also, the relationship between the proposed GMI and other types of SSL (decorrelation-based, infoNCE) are clearly illustrated.
3. Extensive experiments are conducted while demonstrating the effectiveness of proposed GMI, especially, the efficiency comparison is surprisingly, both of the training time and memory cost are significantly decrease with the usage of GMI.

**Weaknesses:**

1. The main contribution of this paper is the finding of tractable surrogate for maximizing mutual information under the Gaussian assumption, resulting in an efficient SSL method. The proposed method demonstrates effectiveness and efficiency on datasets that adhere to or approximate a Gaussian distribution. Although the author mentions that the method maintains performance under non-Gaussian conditions, this claim is based on datasets that did not pass the hypothesis testing for Gaussian distribution. While Fig. 6 shows some similarity between the dataset distribution and Gaussian distribution, the slight disparity is unlikely to significantly impact model performance. Thus, this claim might be over-sell, additional evidence with alternative datasets would be valuable.

2. The results in Table 2 demonstrate a notable enhancement in the efficiency of the proposed method. The author attributes this improvement to the reduced computational cost of mutual information estimators. However, there seems to be a similarity in the calculation process between infoNCE and the proposed GMI outlined in Algorithm 3. Therefore, it is necessary to provide further clarification on the underlying reason for the efficiency difference between the two methods. Additional explanation would be beneficial in addressing this question.

3. Given that the objective of SSL methods is to acquire a discriminative and robust representation, it is important to include visualizations of the learned representations and conduct comparisons. Additionally, it is crucial to further investigate whether the generalization performance of the representation is influenced by the Gaussian assumption. These aspects require additional verification to provide a comprehensive evaluation of the proposed method.

**Questions:**

Please refer to weaknesses

---

> ### Author Response · Authors · 2023-11-19
> **Response to Reviewer r9Qc**
>
> Thank you for your positive recognition of our work and valuable feedback!
>
>
> **Question 1:** Although the author mentions that the method maintains performance under non-Gaussian conditions, this claim is based on datasets that did not pass the hypothesis testing for Gaussian distribution. While Figure 6 shows some similarity between the dataset distribution and Gaussian distribution, the slight disparity is unlikely to significantly impact model performance. Thus, this claim might be over-sell, additional evidence with alternative datasets would be valuable. Besides, it is crucial to further investigate whether the generalization performance of the representation is influenced by the Gaussian assumption.
>
> **Answer:** In our revision, we visualize the histograms for three additional datasets: Ogbn-Arxiv, CS, and Computers. On the CS and Computers, the representations deviate noticeably from the Gaussian distribution, but our proposed method still exhibits satisfactory performance on these two datasets. This can serve as empirical evidence for the effectiveness of our method in non-Gaussian scenarios.
>
> Below, we analyze from the perspective of multi-view self-supervised learning why our method remains effective in non-Gaussian scenarios. Current multi-view self-supervised learning methods can be decomposed into two complementary components: a) enhancing the consistency between representations from different views, and b) employing specific strategies to avoid collapsed solutions. The former usually brings representations from different views closer in the representation space. Among existing methods, the latter includes strategies such as negative sampling (GRACE, SimCLR, and MVGRL), decorrelation strategies (CCA-SSG and Barlow Twins), and asymmetric network architectures (BYOL). In our paper, we demonstrate that our method can indeed achieve both effects a) and b) in Sections 3.2 and 5.1, and this analysis is not conducted under the Gaussian prior, which is a crucial factor contributing to the effectiveness of our approach in non-Gaussian scenarios.
>
> If our method indeed exhibits suboptimal performance in non-Gaussian scenarios in practical applications, a coping strategy can be to impose probabilistic constraints on the representations to encourage them to approximate a Gaussian distribution. For a representation matrix $\mathbf{H} \in \mathbb{R}^{N \times d}$ from view A or B, we first calculate mean $u_i$ and variance $v_i$ of $N$ points from the $i$-th dimension. Then, a 1-dim Gaussian distribution $p_i(x)$ based on $u_i$ and $v_i$ can be constructed. Further, we calculate the product of probabilities of $N$ samples in the $i$-th dim from $p_i(x)$, that is, $\prod_{k=1}^{N} p_i(H_{ki})$. In practice, we utilize its logarithm $\sum_{k=1}^{N} \log p_i(H_{ki})$. Considering all dimensions, we construct a loss $\mathcal{L}\_{gau} = - \sum_{i=1}^{d} \sum_{k=1}^{N} \log p_i(H\_{ki})$ and make representations tend towards a Gaussian distribution by minimizing $\mathcal{L}\_{gau}$. It can be realized by attaching $\mathcal{L}_{gau}$ to the original loss. After applying the above constraints, we observed that the performance nearly did not improve on the experimental datasets. The reason is that our method is not influenced by the non-Gaussian conditions on these datasets.
>
> **Question 2:** It is necessary to provide further clarification on the underlying reason for the efficiency difference between the infoNCE and the proposed GMI.
>
> **Answer:** In addition to the objective function, our method and InfoNCE-based methods also differ significantly in the overall network architecture. Our architecture only comprises one encoder, which maps the input space to the representation space. In contrast, InfoNCE-based methods include an additional projection head following the encoder, which maps node representations to a new embedding space, adding computational burden and decreasing efficiency. In addition, we directly compute the loss function in the representation space while InfoNCE-based methods operate in the new embedding space. Compared to InfoNCE-based methods, our encoder parameters are closer to the source of gradients, allowing for better optimization and, consequently, faster convergence.
>
> **Question 3:** Given that the objective of SSL methods is to acquire a discriminative and robust representation, it is important to include visualizations of the learned representations and conduct comparisons.
>
> **Answer:** Thank you for your suggestions. In Appendix H.6 of our revision, we have included t-SNE visualizations of our method under different configurations and compared them with other methods, offering a more profound understanding of our approach.
>
>
>
> The above is our response to the questions in your review. Once again, we appreciate your valuable comments and suggestions.

---

> > ### Comment · Reviewer_r9Qc · 2023-11-21
> > **reply to author's response**
> >
> > Thanks for the author's response. My concerns are well addressed, so I would like to maintain my score.

---

> > > ### Author Response · Authors · 2023-11-22
> > > **Response to Reviewer r9Qc**
> > >
> > > Thank you for your response and the recognition of our work!

---

### Official Review · Reviewer_qTvu · 2023-10-31

**Soundness:** 2 fair
**Presentation:** 2 fair
**Contribution:** 2 fair
**Rating:** 5
**Confidence:** 3

**Summary:**

The paper discussed self-supervised learning based on two views of data on graphs using decorrelation of the embedding dimensions to avoid collapse. The decorrelation is motivated under a Gaussian assumption of node embedding (but no Gaussian prior is actually imposed). The specific objective focuses on the log determinant form of entropy and maximizes this entropy with the self-supervised penalty that two augmented views of the same data should have similar views. This form is shown to correspond to information maximization.  Besides these results, empirical evidence highlighting the superiority of the method over other methods including directly supervised on a number of standard node classification problems for data on graphs (publication graphs, etc.).

**Strengths:**

The methods in the paper have a sound basis and are logical (the Gaussian assumption approach seems well-motivated, since the embedding network is being learned). The paper is mostly clear, and most formulations are clearly stated mathematically. The self-supervised learning is still of significant interest to the community. The reported results are consistently better.  The results in the appendix show a variety of tests that verify the quality of the method.

**Weaknesses:**

**Clarity of context**
The paper is not clear on the graph context. Although many references to related work, the formulation of the problem in graph terms is missing. At the same time it is not clear why the paper should be limited in scope to graph data. While the structure of graph data is interesting, the data itself at nodes may be simpler than data say from images? I don't see a strong reason to limit the scope to data on nodes with graph now. In any case, it should be stated clearly upfront, but until the contributions (bottom of page 2) the setting of graphs is not clear. The reader is left to ponder if this data on graphs, are the nodes the data, or is the graph the data?  Even at the end of the introduction, it is not exactly clear what is envisioned. It is not until experiments that the graph convolutional network (GCN) is mentioned.  At the end of section 3.1 it should be stated the domain, range, and operation of $f_\theta$ as a graph convolutional encoder.

**More discussion of uncorrelated Gaussians in auto-encoders, blind source separation, etc.**
The Gaussian assumption approach seems well motivated since it is not about the data, but rather about the embedding of data. Yet, I was expecting to see more references to non-linear independent component analysis and disentanglement methods for the latent space of auto-encoders as in beta-VAE. Foundational references to decorrelation methods for independent component analysis could also be mentioned.

The derivation of mutual information for Gaussians is not novel and is well-known. The statement that this is a contribution (and a proposition that requires a proof) is misleading.

**Organization of results in not straightforward**
Property 2 is a statement of PCA, I'm not sure I understand the reasoning of "Property 2 potentially suggests that the unevenness of the eigenvalues of the covariance matrix leads to the issue of dimensional collapse." Theorem 1 is talking about an empirical version but could be combined with Proposition 2. It seems easier to say that maximizing the determinant ensures maximum entropy and isotropic covariance.

**Major concerns**
Concern 1).  It is not clear why the method is not directly compared with the work by Ozsoy et al. in the main body. As this work mentions in an appendix, this work already developed decorrelation-based self-supervision and has a similar objective.

[2]Ozsoy, Serdar, Shadi Hamdan, Sercan Arik, Deniz Yuret, and Alper Erdogan. "Self-supervised learning with an information maximization criterion." Advances in Neural Information Processing Systems 35 (2022): 35240-35253.

Concern 2: rigorous and fair comparison for hyper-parameters )
How are hyper-parameter searched? What are the details of the linear classifier, as many are possible, i.e., regularization or penalty hyper-parameters and how are these selected? To be truly self-supervised a validation set would not be available.  Is it a grid search on dimension and trade-off parameter?  The augmentation intensities $p_f$ and $p_e$ have to be selected and the results in Figure 10 show that these also have a large effect. Notably, the variation across hyper-parameters seems greater than the performance gains of the proposed methodology. CCA-SSG outperforms GMIM and GMIM-IC (with a hyper-parameter) outperforms CCA-SSG.

Are competing baselines using defaults or do they also enjoy a hyper-parameter search? It is not clear if the hyper-parameter search is fair: "to get relevant results based on the officially released source code. " This does not clarify if hyper-parameter selection is used for all methods.

**Minor points:**
Definition 1 should be specific that the densities are required (not the distribution as stated).

The discussion of Proposition 3 about distributions on hyper-spheres can be related to Gaussian distribution [1].
[1] Davidson, Tim R., Luca Falorsi, Nicola De Cao, Thomas Kipf, and Jakub M. Tomczak. "Hyperspherical variational auto-encoders." In 34th Conference on Uncertainty in Artificial Intelligence 2018, UAI 2018, pp. 856-865. Association For Uncertainty in Artificial Intelligence (AUAI), 2018.

'obey a potential Gaussian distribution' -> 'obey a Gaussian distribution'

Page 1 not clear what is meant by "specific equipment"

Page 2"decorrecting differences"

"Imposing cross-view identity constraint" -> "Imposing a cross-view identity constraint"
I'm not sure what "as a pivot to elucidate" means.

Page 5 "Th" in between equations 4 and 5.

After equation (7), "still stands up" is too informal. The point is a similar relation holds for $B$.

Page 7 "guilds the model" -> "guides the model"

Nitpick : trace can be denoted in Roman font $\mathrm{tr}$ instead of italics just like $\ln$ and $\det$.

**Questions:**

Please see questions above regarding comparison to previous work and hyper-parameter search.

---

> ### Author Response · Authors · 2023-11-19
> **Response to Reviewer qTvu (Part 1)**
>
> Thank you for your thorough review and constructive feedback!
>
>
>
> **Question 1:** The paper is not clear on the graph context. Why is the paper limited in scope to graph data? Besides, the introduction part does not specify whether the data is the graph or the nodes.
>
> **Answer:** As stated in Appendix E, visualized histograms of *node representations* and the Gaussian assumption provided the initial motivation for our research, so we naturally regard (node representations on) graphs as objects. We believe that our approach can be extended to image data, but visual self-supervision typically relies on significant computational resources. Regrettably, the resources available to us do not permit us to conduct relevant experimental studies. This point was also highlighted in the *Limitations* section of the initial submission.
>
> Thank you for your reminder about the unclear description. The data is nodes on the graph. In our revised version, we have employed the terms such as "node-level" and "node representations" at multiple places to indicate this point. Additionally, we have added more descriptions of crucial issues such as the composition of the encoder.
>
> **Question 2:** Why does Property 2 potentially suggest that the unevenness of the eigenvalues of the covariance matrix leads to the issue of dimensional collapse?
>
> **Answer:** We have provided a detailed explanation of this question in the red paragraph on page 18 of our revision.
>
> **Question 3:** The authors should directly compare the proposed method with CorInfoMax [1].
>
> **Answer:** In Table 1 of our revision, we add CorInfoMax as a baseline. When conducting related experiments, we adopt the same encoder architecture for CorInfoMax as ours. CorInfoMax falls behind our method in performance. Our method directly optimizes Gaussian mutual information in the representation space, while CorInfoMax adds an additional projection head following the encoder. This component is somewhat redundant and hinders the direct optimization of representations.
>
> **Question 4:** Some questions and issues regarding comparison to previous works and experimental settings.
>
> (1) To be truly self-supervised learning, a validation set would not be available. Is a grid search employed for hyper-parameter selection via a validation?
>
> *Answer:* Based on a validation set in the downstream node classification task, we conduct a grid search for hyper-parameters of the proposed method in our experiments. In our paper, all self-supervised baselines go through such a process. *During the self-supervised pretraining process, no labels are involved in the training.*
>
> Indeed, as you mentioned, truly self-supervised learning should not have access to a validation set. However, in most existing literature including classical works in both **vision and graph fields** (e.g., MAE, GRACE, and CCA-SSG), a hyper-parameter search is performed based on a validation of downstream task. In other words, they also do not achieve truly self-supervised learning. We think that this issue is primarily due to the huge gap between self-supervised proxy tasks and downstream tasks such as classification in the graph and vision fields. A well-performed self-supervised proxy task (e.g., a small InfoNCE loss) does not guarantee that the pre-trained model will still perform well on downstream classification tasks. Therefore, it is often necessary to perform hyper-parameter search based on the validation set of downstream tasks. Without relying on a validation set, constructing specific metrics or strategies to measure the effectiveness of self-supervised pretraining remains an important under-explored research problem. This is also a significant reason why the research on self-supervised learning in the fields of vision and graph lags behind that in the NLP. In NLP, proxy tasks like masked language modeling align well with downstream applications such as question-answering. Generally speaking, the issue the reviewer points out is actually one that the entire self-supervised learning community for graph and vision is facing and actively seeking solutions for.

---

> ### Author Response · Authors · 2023-11-19
> **Response to Reviewer qTvu (Part 2)**
>
> (2) How are hyper-parameter searched? What are the details of the linear classifier?
>
> *Answer:* The hyper-parameters are searched by a grid search as follows:
>
> representation dimension: [64, 128, 256, 512]
>
> $p_f$: [0, 0.1, 0.2, 0.3, 0.4, 0.5, 0.6]
>
> $p_e$: [0, 0.1, 0.3,  0.5, 0.6, 0.8, 1.0]
>
> trade-off factor $\beta$: [0.5, 1, 2, 3, 4, 5, 7, 10]
>
> The reason for the coarser search in $p_e$ is that existing studies (e.g., GRACE and CCA-SSG) have already demonstrated that graph self-supervised learning is less sensitive to structural augmentation. It is worth noting that when conducting sensitivity analysis for specific hyper-parameters, in order to make the experiments more thorough, the hyper-parameter settings will have a higher granularity than grid search.
>
> After self-supervised pre-training, we can obtain node representations $\mathbf{H} = [\mathbf{h}_1,...,\mathbf{h}_N]^{\top}$ and evaluate their qualities with simple linear classifier $y = \mathbf{W} x + \mathbf{b}$, where $\mathbf{W}$ and $\mathbf{b}$ denote learnable parameters. The linear classifier realizes mapping from representation space into label space and is used to assess the linear separability of representations.
>
> (3) The augmentation intensities $p_f$ and $p_e$ have to be selected and the experimental results show that these also have a large effect.
>
> *Answer:* Indeed, the augmentation intensity has a great influence on performance, which have been explored in a previous work [3]. In fact, we can observe that, when $p_f$ in [0.2, 0.4] and $p_e$ in [0.0, 0.6], the performance keep relatively stable and good.
>
> (4) Are competing baselines using defaults or do they also enjoy a hyper-parameter search?
>
> *Answer:* Based on the official codes of the baselines, we perform a hyper-parameter search to obtain their best results. For instance, in the case of GRACE, the official code is used to conduct experimental evaluations on random splits of Cora, Citeseer, and Pubmed. The given hyper-parameters perform poorly on the public splits, and we obtained the reported results in our paper after conducting a hyper-parameter search.
>
> **Question 5:** In terms of writing, there are some typos and issues.
>
> **Answer:** Thank you for your careful reading and detailed comments very much! We have corrected the typos and issues in our revised version based on your feedback. Here are a few points that need special clarification.
>
> (1) The discussion of Proposition 3 about distributions on hyper-spheres can be related to Gaussian distribution.
>
> *Answer:* Thank you for your suggestions! However, while preparing this rebuttal, we have not fully comprehended how the data points adhere to a Gaussian distribution on the hypersphere. For example, in the case of a two-dimensional space, the data points on a unit circle appear to be unable to conform to a Gaussian distribution. The paper [2] abandons the commonly used Gaussian prior and instead turns to a hyperspherical latent space. If you have any further suggestions or ideas, please do not hesitate to tell us.
>
> (2) What does "as a pivot to elucidate" means?
>
> *Answer:*  GMIM-IC is a contrastive objective derived from maximizing Gaussian Mutual Information. Here, "as a pivot to elucidate" means that we will demonstrate the relationship between contrastive methods and decorrelation-based methods using GMIM-IC as an *intermediary*.

---

> ### Author Response · Authors · 2023-11-19
> **Response to Reviewer qTvu (Part 3)**
>
> **Question 6:** More discussions of uncorrelated Gaussians in auto-encoders, independent component analysis, and blind source separation are expected.
>
> **Answer:** Our method expects to enforce the representations to achieve an isotropic Gaussian distribution with the aim of decoupling various dimensions and learning diverse representations.
>
> Conventional Independent Component Analysis (ICA) or Blind Source Separation (BSS) thinks that the observed mixed signals are obtained through a linear combination of source signals and aims to recover latent variables from observations. The traditional ICA assumes that the source signals follow non-Gaussian distributions and are statistically independent. The assumption of statistical independence among latent variables is, in fact, a decoupling or independence constraint. Non-linear ICA posits that the observed signals are obtained through a nonlinear transformation of the source signals.
>
> In variational auto-encoders (VAE), there is a term used to minimize the KL divergence between the variational posterior and the prior distribution. The chosen prior distribution is typically selected to satisfy certain independent properties, such as an isotropic Gaussian distribution. As a result, the KL divergence term potentially imposes a decoupling constraint on the latent variables. The $\beta$-VAE [4] multiplies the KL divergence term by a coefficient to enhance the decoupling effect on the latent variables. The KL divergence term and the entropy maximization term in our paper share a similar underlying principle. Similar to decorrelation-based self-supervised learning methods such as CCA-SSG, DIP-VAE [5] directly regularizes the elements in the covariance matrix of the posterior distribution, making it approach the identity matrix. Generally speaking, research on disentangled representation learning in VAEs mainly focuses on improving the KL divergence term or finding its alternatives.
>
> We are currently organizing the content related to this question. We are conducting more extensive research and discussions, and  will integrate the relevant content into our revision.
>
>
>
> [1] Self-supervised learning with an information maximization criterion. NeurIPS 2022.
>
> [2] Hyperspherical variational auto-encoders. UAI, 2018.
>
> [3] What makes for good views for contrastive learning? NeurIPS, 2020.
>
> [4] beta-vae: Learning basic visual concepts with a constrained variational framework. ICLR, 2016.
>
> [5] Variational inference of disentangled latent concepts from unlabeled observations. ICLR, 2018.
>
>
>
> Thank you again for your detailed and valuable comments. The above is our response to your questions, and we are looking forward to further discussions with you.

---

> > ### Comment · Reviewer_qTvu · 2023-11-20
> >
> > I want to thank the authors for replying to many of my questions or points. I do think the authors may have misunderstood my misunderstandings not in what they were saying (uneven eigenvalues correspond to non-isotropic distributions), but the rather my confusion why this was even suprising.
> >
> > Nonetheless, I think my initial score was a fair assessment of contribution of work. I agree with the sentiments of reviewer jkcF in terms of weaknesses, and am not convinced that this contribution to SSL is substantial, even in the context of SSL for graph data.
> >
> > Note that Cover & Thomas's famous "Elements of Information Theory" is a reference with mutual information for Gaussians.

---

> > > ### Author Response · Authors · 2023-11-20
> > > **Discussion with Reviewer qTvu**
> > >
> > > Dear Reviewer qTvu,
> > >
> > > Thank you very much for your response.
> > >
> > > Firstly, we are sorry for having misunderstandings regarding some of your initial reviews. Indeed, the statement regarding "the unevenness of eigenvalues potentially leads to dimensional collapse issue" is not a new concept and has been well-known to the reviewer. However, in order to maintain the self-completeness of our article and facilitate a smoother understanding of the conclusions presented in our paper by readers, the relevant descriptions in Section 5.1 still hold value and significance and would be retained after modification. Specifically, to streamline and clarify the relevant statements, we have removed Theorem 1 in our new revision and reorganized our statements of Section 5.1 in a more natural manner based on your comments in "**Organization of results is not straightforward**".
> > >
> > > Thank you for providing the reference. In our revision, we have cited the article "Elements of Information Theory". After a careful browsing, we find that this paper contains the derivation of Gaussian entropy, which is crucial for deriving Gaussian mutual information. To be honest, when deriving the Gaussian mutual information, we did not find a complete derivation process in the existing literature. The derivations presented in the paper were conducted independently by us, which consumed a significant amount of efforts on our part.
> > >
> > > Lastly, please allow us to restate the contributions of our paper. On the one hand, in most existing methods, the presence of negative samples and additional projection heads adds redundancy and complexity of self-supervised learning (SSL) methods. Therefore, we aim to simplify SSL approaches under some specific settings. Specifically, under the Gaussian assumption, we propose a SSL method based on the feasible closed-form formula of mutual information. The advantage in efficiency is a notable strength that should not be overlooked. On the other hand, it is an important issue to establish theoretical connections and a unified perspective for different self-supervised learning methods, which has received relatively limited research attention. Our study contributes to this aspect and provides insights into bridging various self-supervised learning methods.
> > >
> > > We would also like to express our gratitude to the other two reviewers for their efforts. We have made our best to address the weaknesses pointed out by the reviewer jkcF and eagerly await his feedback.
> > >
> > > We sincerely appreciate your valuable comments and active discussion. Based on your feedback, we have further improved our paper and sincerely look forward to further discussion with you.

---

> ### Author Response · Authors · 2023-11-23
> **The added discussion aoubt auto-encoders and independent component analysis**
>
> Dear Reviewer qTvu,
>
> We provide a review and discussion of independent component analysis (ICA) and variational autoencoders (VAEs) from the perspective of disentangled representation learning:
>
> Our method expects to enforce the representations to achieve an isotropic Gaussian distribution with the aim of decoupling various dimensions and learning diverse representations, which is closely linked to a branch of deep learning called Disentangled Representation Learning (DRL) [1, 2, 3]. The objective of disentangled representation learning is to achieve a clear separation of the distinct, independent, and informative generative factors inherent in the data [3]. DRL emphasizes the statistical independence among latent variables, which can be traced back to Independent Component Analysis (ICA) [4].
>
> Independent Component Analysis, a computationally efficient Blind Source Separation (BSS) [5] technique, thinks that the observed mixed signals are obtained through a linear combination of source signals and aims to recover latent variables from observations. The traditional ICA assumes that the source signals follow non-Gaussian distributions and are statistically independent. The assumption of statistical independence among latent variables is, in fact, a disentangled or independence constraint. Non-linear ICA [6] posits that the observed signals are obtained through a nonlinear transformation of the source signals.
>
> The variational autoencoder (VAE) [7] is a modification of the autoencoder that incorporates the concept of variational inference, which can realize dimension-wise disentanglement. In VAEs, there is a term used to minimize the KL divergence between the variational posterior and the prior distribution. The chosen prior distribution is typically selected to satisfy certain independent properties, such as an isotropic Gaussian distribution. As a result, the KL divergence term potentially imposes a independent constraint on the latent variables. The $\beta$-VAE [8] multiplies the KL divergence term by a penalty factor $\beta$ to enhance the disentangling effect on the latent variables. The KL divergence term shares a similar underlying principle with the entropy maximization term in our paper. Chen et al. [9] demonstrate that the penalty term tends to enhance the dimension-wise independence of the latent variable, but it also diminishes the capacity of the latent variables to preserve information from the input. Similar to decorrelation-based self-supervised learning methods such as CCA-SSG, DIP-VAE [10] directly regularizes the elements in the covariance matrix of the posterior distribution, making it approach the identity matrix. FactorVAE [11] introduces a term known as *Total Correlation* to quantify the level of dimension-wise independence.
>
>
>
> The above content has been incorporated into Appendix F of our revised version and will continue to be enriched in the following days. Thank you for your time and efforts during the review process!
>
>
>
>
>
> [1] On Causally Disentangled Representations.
>
> [2] Weakly Supervised Disentangled Generative Causal Representation Learning.
>
> [3] Representation Learning: A Review and New Perspectives.
>
> [4] Independent component analysis: a tutorial introduction.
>
> [5] Blind source separation.
>
> [6] Nonlinear independent component analysis.
>
> [7] Auto-Encoding Variational Bayes.
>
> [8] beta-{VAE}: Learning Basic Visual Concepts with a Constrained Variational Framework.
>
> [9] Isolating Sources of Disentanglement in Variational Autoencoders.
>
> [10] Variational inference of disentangled latent concepts from unlabeled observations.
>
> [11] Disentangling by Factorising.

---

### Meta-Review · Area_Chair_9yui · 2023-12-06

**Metareview:**

This paper originates from the Gaussian assumption of graph node embeddings and proposes to maximize the mutual information (MI) between two augmented views to perform graph self-supervised learning (SSL). The author's approximation of the MI has better stability. It gives the best overall embedding based on subsequent classification tasks against a wide array of baselines.

*Strengths*:

- The method is easy to follow and makes intuitive sense as illustrated by the diagram.

- The authors made many efforts in the experimental design and the performance is consistent and thoroughly tested.

Weaknesses:

- There is not enough effort in positioning this work in the literature. The authors' comparison against CorInfoMax should be based on technical arguments in the main text. The difference in the encoder architecture (projection heads), or the motivation, is not a convincing argument on how it advances over the SOTA.

- The formal statements can be improved and the quality of these theoretical parts is below the publication standard. For example, based on reviewer jkcF, the closed-form Gaussian mutual information is useful, as not everyone in the community knows it, but is common knowledge and should not be a novel contribution. Overall, some of these statements can be moved into the appendix. Some can be better (more formally and aligned with the proposed MI approximation) stated.

- The reviewers suggested additional empirical study, sensitivity wrt the backbone encoder, extension to image datasets, and semi-supervised evaluations, are not sufficiently addressed in the rebuttal.

**Justification For Why Not Higher Score:**

There has been a thorough discussion between the authors and the reviewers. Based on the weaknesses raised, this work should go through a major revision to clarify its technical improvement over the literature and to improve the quality of the writing.

**Justification For Why Not Lower Score:**

N/A

---

### Decision · Program_Chairs · 2024-01-16

Reject